# Effects of Nitrogen and Phosphorus Addition on Agronomic Characters, Photosynthetic Performance and Anatomical Structure of Alfalfa in Northern Xinjiang, China

**Yanliang Sun** [ID]**, Xuzhe Wang, Chunhui Ma and Qianbing Zhang** *

College of Animal Science and Technology, Shihezi University, Shihezi 832003, China; yanliangsun820@gmail.com (Y.S.); wangxuzhe@shzu.edu.cn (X.W.); chunhuima@126.com (C.M.)
* Correspondence: qbz102@shzu.edu.cn

**Abstract:** The productivity of alfalfa is associated with a large amount of nitrogen (N) and phosphorus (P); the addition of exogenous N and P fertilizers can fully exploit the growth potential of alfalfa. However, there is uncertainty about the relationship between changes in alfalfa productivity and photosynthetic physiology and anatomy. We conducted field fertilization experiments on alfalfa in the second and third years under drip irrigation, as well as measurement of the photosynthetic physiology, anatomical structure and agronomic traits of alfalfa at different levels of N (0, 120 kg·ha$^{-1}$) and different levels of $P_2O_5$ (0, 50, 100 and 150 kg·ha$^{-1}$). The results showed that the dry matter yield (DMY), crude protein (CP), net photosynthetic rate ($P_n$) and specific leaf weight (SLW) were increased by 2.10~11.82%, 4.95~11.93%, 4.71~7.59% and 2.02~7.12% in the N application treatment compared with the non-N application treatment, while the DMY, CP, $P_n$ and SLW were increased by 3.19~17.46%, 1.99~8.42%, 6.15~24.95% and 2.16~11.90% in the P application treatment compared with the non-P application treatment. N and P increase the thickness of the spongy tissue (ST) of alfalfa, which will facilitate the entry and exit of gas and water, and will further affect the photosynthetic indexes, such as stomatal conductance ($G_s$) and transpiration rate ($T_r$), of alfalfa leaves. Increased palisade tissue (PT) thickness will also enhance the adaptability of plant leaves to strong sunlight, thereby increasing the maximum net photosynthetic rate ($P_{max}$) and light saturation point (LSP). Fertilization treatment showed the highest utilization efficiency for low light and better adaptation to strong light, but the $R_d$ decreased. The comprehensive scores of principal component analysis for anatomical structure, photosynthetic performance and agronomic traits were $N_1P_2 > N_0P_2 > N_1P_3 > N_1P_1 > N_0P_3 > N_0P_1 > N_1P_0 > N_0P_0$. Therefore, the application of N and P fertilizers contributed to the adaptive changes in alfalfa leaf anatomy and the improvement of photosynthetic capacity, which were beneficial to the improvement of alfalfa dry matter yield, growth traits and nutritional quality, with the most obvious improvement effect obtained with the application of 120 kg·ha$^{-1}$ of N and 100 kg·ha$^{-1}$ of $P_2O_5$.

**Keywords:** alfalfa; N and P addition; anatomical structure; photosynthetic performance; agronomic traits

## 1. Introduction

Alfalfa (*Medicago sativa* L.) is a perennial leguminous plant with high yield, rich nutrition and strong adaptability [1,2]. Soil available nitrogen (AN) and available phosphorus (AP) deficiency are some of the most important factors limiting the productivity of alfalfa [3]. Alfalfa growth requires a large amount of N; the N fixation of rhizobia converts molecular N in the air into ammonium N that can be absorbed and utilized by alfalfa [4]. However, there is still considerable uncertainty about whether alfalfa needs additional N fertilizer [5,6]. Measuring the effect of N fertilization on alfalfa growth must take into account rhizobial activity, soil microenvironment and fertility and environmental factors. P has been shown to significantly improve alfalfa yield and quality, but its effect varies

depending on the level of available P in the soil. P is abundant in soil, but the majority of it is difficult to obtain due to the low solubility of inorganic forms of P in water, resulting in P concentrations of 10 μm or less in soil solution and poor soil P mobility [7,8].

Plant leaves are the major location of photosynthesis, which serves as the center for energy conversion in primary producers. Simultaneously, they are more sensitive to environmental and soil changes, and alterations in their epidermal tissue, palisade tissue, spongy tissue and other leaf anatomical features are critical for photosynthesis [9]. N has been demonstrated to promote the division and development of plant leaf palisade cells as well as the formation of sponge cells, particularly during the middle and late phases of plant growth, and this boosting impact progressively rises as the N concentration increases [10]. Under the condition of leaf N deficiency, the addition of appropriate N fertilizer may assist in enhancing plant light energy use efficiency, promoting plant growth and increasing above-ground biomass [11]. Alfalfa is a plant that requires a lot of P and is very sensitive to both an abundance and deficiency of P [12,13]. Studies have shown that without P fertilizer, the absorption and utilization of alfalfa will result in P deficiency in the soil, which in turn will cause a halt in alfalfa growth. In contrast, excessive P application results in a surplus of soil P, leading to P accumulating in the soil [14]. When the soil is deficient in P, the radial thickening of the vascular column in the fine root anatomy, the degree of lignification and the number of ducts increase significantly, thereby increasing the transport capacity of the plant root system. Simultaneously, the cells in the root endothelium disintegrate and release dephosphorylated elements, relieving the plant of its stress caused by the P-deficient environment. By contrast, as a result of P stress, the thickness of the spongy and palisade tissues inside plant leaf tissue tends to diminish [15]. Meanwhile, the strength of plant photosynthesis directly affects its dry matter accumulation, soluble sugar and starch as the main products of photosynthesis; when P stress affects plant photosynthesis, this will also indirectly affect the synthesis of soluble sugar and starch in leaves [16].

Although some progress has been made in research on the production performance of alfalfa under different fertilization conditions, no research has paid attention to the deeper reasons for the high dry matter yield, good growth traits and good nutritional quality of alfalfa. Here, we tested the following three hypotheses: (1) N and P fertilization can improve alfalfa productivity by increasing photosynthetic efficiency and photosynthetic area; (2) N and P fertilization promotes anatomical structure, leading to a wider range of light intensity adaptive changes, promotes photosynthetic pigment production and further enhances photosynthesis; and (3) alfalfa growth performance and nutrient supply are intimately connected, and N and P fertilizer treatments increase dry matter production and nutritional quality compared to those without fertilization.

## 2. Materials and Methods

### 2.1. Experimental Site

The field experiments were conducted at the Modern Water-Saving Irrigation Corp's Key Laboratory Test Base in Shihezi (44°20′ N, 86°30′ E), Xinjiang province, China, in 2020 and 2021. Shihezi is located in the arid regions in the northwest part of China, where agriculture is mainly based on drip irrigation water-saving techniques with integrated water–fertilizer. The average annual temperature is 7.5 °C, the precipitation is 225 mm, the sunshine duration is 2770 h, and the evaporation is 1250 mm, and gray desert soils dominate the experimental area. The soil tillage layer (0~20 cm) contained 21.56 g·kg$^{-1}$ organic matter, 1.18 g·kg$^{-1}$ total nitrogen, 0.53 g·kg$^{-1}$ total phosphorus, 19.30 mg·kg$^{-1}$ available phosphorus, 145.47 mg·kg$^{-1}$ alkaline nitrogen, 119.80 mg·kg$^{-1}$ available potassium and pH = 7.95. The daily (10:00–20:00) average photosynthetically active radiation during the photosynthesis measurement in 2020 was 1574.17 μmol·m$^{-2}$·s$^{-1}$ and maximum photosynthetically active radiation was 1998.02 μmol·m$^{-2}$·s$^{-1}$. The average field $CO_2$ concentration during the photosynthesis measurement was 405.52 μmol·mol$^{-1}$, the relative air humidity was 24.59%, the atmospheric temperature was 33.19 °C, and leaf surface saturated vapor pressure was 3.40 kPa. In 2021, the daily average photosynthetically active

radiation during the photosynthesis measurement was 1587.89 $\mu mol \cdot m^{-2} \cdot s^{-1}$ and maximum photosynthetically active radiation was 1989.24 $\mu mol \cdot m^{-2} \cdot s^{-1}$; the average field $CO_2$ concentration during the photosynthesis measurement was 401.54 $\mu mol \cdot mol^{-1}$, the air relative humidity was 25.19%, the atmospheric temperature was 34.27 °C, and the leaf surface saturated vapor pressure was 3.39 kPa.

### 2.2. Experimental Design and Crop Management

A two-factor randomized complete block design with three replicates was implemented in each treatment. The experimental treatments were the factorial combinations of two gradients of N fertilizer (as N equivalent; 0 ($N_0$), 120 $kg \cdot ha^{-1}$ ($N_1$)), and four gradients of P fertilizer (as $P_2O_5$ equivalent; 0 ($P_0$), 50 ($P_1$), 100 ($P_2$) and 150 ($P_3$) $kg \cdot ha^{-1}$)—a total of 8 treatments were applied with N and P interaction. The N and P addition trials were conducted for two consecutive years, during which no rhizobia inoculation was carried out. The N fertilizers used in the experiment were urea (containing N 46%) and mono-ammonium phosphate (containing $P_2O_5$ 11%). The P fertilizer was mono-ammonium phosphate (containing $P_2O_5$ 52%). $N_0$ and $N_1$ treatments were applied by adding N fertilizer to maintain the same level of N under different P treatments, and fertilizer was applied through a drip irrigation system in the highly efficient water- and fertilizer-saving method of "fertilizer follows water". Fertilizer was applied by a drip irrigation system with water at the branching stage (19 April 2020 and 18 April 2021) and 3–5 days after mowing of the first three cuts (25 May, 4 July, 14 August, 2020, 27 May, 3 July, 4 August 2021).

The WL366HQ alfalfa seeds were sown manually in strips in 2019 at a seeding rate of 18.0 $kg \cdot ha^{-1}$, a row spacing of 20 cm and a sowing depth of 2 cm. Deep ploughing of the land before sowing provides a suitable environment for the deep penetration of alfalfa roots, and timely suppression after sowing is conducive to the close combination of seed and soil, and the moisture retention of the soil surface. The drip irrigation tape was buried shallowly to 10 cm below the soil layer with a spacing of 60 cm. The specifications of each plot were 4 m × 6 m, and a 1 m protection row was set between the plots to prevent the infiltration of nutrients. In addition to fertilization, other management measures such as watering, weeding and insect control were carried out uniformly following local high-yielding alfalfa field production habits.

### 2.3. Sampling and Measurements

#### 2.3.1. Agronomic Traits

Dry matter yield (DMY) was measured by the sampling method at the early flowering stage of alfalfa (flowering 5~10%). Meanwhile, plant height (PH) stem diameter (SD) and stem to leaf ratio (S/L) were measured.

Nutritional quality: Total nitrogen content was determined using the Kjeldahl method, and total crude protein (CP) was calculated by multiplying the obtained results by 6.25; the neutral detergent fiber (NDF) and acid detergent fiber (ADF) content were determined by the Van Soest method, and the ether extract (EE) was determined by the ether extraction method.

#### 2.3.2. Photosynthetic Indexes

Sunny and cloudless weather was selected for field photosynthesis measurements using an LI-6400 gas exchange device and a chlorophyll fluorescence induction system (LI-COR Inc., Lincoln, NE, USA) from 11:00 a.m. to 13:00 a.m., in the first flowering of the second cut of alfalfa (26 June 2020 and 28 June 2021). The measurement indexes included net photosynthetic rate ($P_n$), intercellular $CO_2$ concentration ($C_i$), stomatal conductance ($G_s$) and transpiration rate ($T_r$); meanwhile, environmental factors such as photosynthetically active radiation (PAR), field $CO_2$ concentration ($C_a$), relative air humidity (RH), atmospheric temperature ($T_a$) and leaf surface saturated vapor pressure ($V_{pdl}$) were recorded. The

calculation formulas for instantaneous water use efficiency (WUE) and stomatal limit value ($L_s$) are as follows:

$$\text{WUE} = P_n / T_r \tag{1}$$

$$L_s = (C_a - C_i)/C_a \tag{2}$$

We used the LI-6400 portable photosynthesis instrument and the LI-6400-02B red and blue light source to measure the light response parameters of the leaves. The leaves were induced under 1200 $\mu\text{mol}\cdot\text{m}^{-2}\cdot\text{s}^{-1}$ light intensity for 20 min before measurement. The photosynthetically active radiation (PAR) gradient was set to 2000, 1500, 1000, 800, 600, 400, 100, 50, 20 and 0 $\mu\text{mol}\cdot\text{m}^{-2}\cdot\text{s}^{-1}$. Maximum net photosynthetic rate ($P_{max}$), light compensation point (LCP) and light saturation point (LSP) were calculated by fitting photosynthetic data using the C3 plant light response model [17], and apparent quantum yield (AQY) and dark respiration rate ($R_d$) were calculated using a response linear fit of the photosynthetic low light intensity below 200 $\mu\text{mol}\cdot\text{m}^{-2}\cdot\text{s}^{-1}$.

### 2.3.3. Photosynthetic Physiological and Biochemical Indexes

Photosynthetic pigment: Chlorophyll content (Chl) and carotenoid content (Car) were determined by the spectrophotometer method [18].

Leaf area: The leaves were brought back to the laboratory, scanned using a scanner to form images, and the leaf area (LA) was calculated using Image Pro Plus software (Media Cybernetics, Silver Spring, MD, USA).

Specific leaf weight (SLW): The whole green mature leaves of alfalfa were cut, and their total area was determined by the above-mentioned leaf area determination method; then, they were dried and weighed, measurement was repeated 3 times, and the average value was taken. Specific leaf weight is the ratio of the dry matter weight of whole alfalfa leaves to leaf area.

### 2.3.4. Anatomical Structure

To analyze the anatomical structure of alfalfa leaves, before cutting the alfalfa, the mature and healthy fresh leaves at the same leaf position (middle leaflet of the trifoliate compound leaf in the 3rd leaf position) were taken before alfalfa mowing and immediately fixed in FFA; after immersion for 12 h, paraffin sections were made. Photographs were taken at $100\times$ magnification using a computer connected to an Olympus BX43F microscope (Olympus Inc., Tokyo, Japan). Leaf total thickness (LT), main vein thickness (MVT), up-epidermal thickness (UE), down-epidermal thickness (DE), palisade tissue thickness (PT) and spongy tissue thickness (ST) were measured using the software tools provided in Image Pro Plus 6.1 and averaged over multiple measurements. The specific formulas for the compaction of leaf tissue (CTR) and porosity of leaf tissue (SR) are as follows:

$$\text{CTR} = (\text{PT}/\text{LT}) \times 100\% \tag{3}$$

$$\text{SR} = (\text{ST}/\text{LT}) \times 100\% \tag{4}$$

### 2.4. Data Analysis

The effects of N and P on the indicators of alfalfa were examined using a two-way (N, P, N $\times$ P) ANOVA for each year, and multiple comparisons were performed using Duncan's method. The assumptions of normality (Shapiro–Wilk test) and homogeneity (Bartlett's test) were tested before subjecting the data to ANOVA.

Structural equation modeling (SEM) was performed using IBM SPSS AMOS 24 (SPSS Inc., Chicago, IL, USA) software to predict the interactions among chlorophyll, palisade tissue, spongy tissue, $P_n$, $T_r$ and DMY, and several model fitness parameters were used to judge the reasonableness of the model, mainly including the ratio of chi-square value to the degree of freedom (CHI/DF < 3), test *p*-value ($p > 0.05$), asymptotic residual mean square and root square (RMSEA < 0.08), goodness of fit index (GFI > 0.9) and value-added

fit number (NFI > 0.9), and when the model met these conditions, it indicated reasonable model fitness.

The C3 plant light response model is:

$$P_n(I) = \frac{\alpha I(1 - \beta I)}{(1 + \gamma I)} - \varepsilon \tag{5}$$

Among them, $I$ is photosynthetically active radiation; $\alpha$, $\beta$, $\gamma$ and $\varepsilon$ are 4 coefficients. We set the initial values of $\alpha$, $\beta$, $\gamma$ and $\varepsilon$ to 0.01, 0.0001, 0.001 and 0.5, respectively, and substituted the light response curve data and initial values into the model equation to calculate the coefficients $\alpha$, $\beta$, $\gamma$ and $\varepsilon$ of the model by the Myquart method.

The calculation formula of the $P_{max}$ is:

$$P_{max} = \alpha(\frac{\sqrt{\beta + \gamma} - \sqrt{\beta}}{\gamma})^2 - \varepsilon \tag{6}$$

The calculation formula of the LCP is:

$$LCP = \frac{\alpha - \gamma\varepsilon - \sqrt{((\gamma\varepsilon - \alpha)^2 - 4\alpha\beta\varepsilon)}}{2\alpha\beta} \tag{7}$$

The calculation formula of the LSP is:

$$LSP = \frac{\sqrt{(\beta + \gamma)/\beta} - 1}{\gamma} \tag{8}$$

The linear fitting equation for a low light intensity–net photosynthetic rate below 200 $\mu mol \cdot m^{-2} \cdot s^{-1}$ is:

$$P_n(I) = \delta I - R_d \tag{9}$$

where $\delta$ is the slope of the straight line fitting equation. When $I = 0$ $\mu mol \cdot m^{-2} \cdot s^{-1}$, $P_n$ is the $R_d$; when $I = 200$ $\mu mol \cdot m^{-2} \cdot s^{-1}$, $\delta$ is the AQY.

## 3. Results

### 3.1. Agronomic Traits

Under the same N application conditions, the DMY, PH, SD and CP of alfalfa tended to increase first and then decrease with increasing P application. In contrast, S/L, NDF and ADF tended to drop initially and subsequently rise as P application increased (Figure 1). The PH, CP and DMY in $P_2$ treatment were significantly greater than those in non-P treatment ($p < 0.05$), but the NDF, ADF and S/L in non-P treatment were significantly lower ($p < 0.05$). In 2020, the EE of the $P_3$ treatment was significantly higher than the other three treatments ($p < 0.05$), and in 2021, the EE of the $P_2$ and $P_3$ treatments were significantly greater than those of the $P_0$ and $P_1$ treatments ($p < 0.05$). Furthermore, although there was no significant difference in SD among the P application treatments in 2020 ($p > 0.05$), the SD for the $P_2$ treatment was significantly greater than the other three treatments in 2021 ($p < 0.05$). The DMY, PH, SD, CP and EE of alfalfa were increased by 3.19~17.46%, 1.14~8.03%, 0.14~15.4%, 1.99~8.42% and 0.53~22.16%, and the S/L, NDF and ADF of alfalfa were decreased by 3.26~11.55%, 1.24~11.57% and 1.44~8.19% in the P application treatment compared with the non-P application treatment. Under identical P application conditions, N application resulted in significantly greater PH and CP than non-N treatment ($p < 0.05$), but non-N treatment resulted in significantly lower NDF and ADF ($p < 0.05$). In 2020, the DMY and SD of the N treatment were significantly greater than those of the non-N treatment under the $P_0$ and $P_3$ treatments ($p < 0.05$), and the S/L of the N treatment was lower than that of the non-N treatment ($p < 0.05$), while the EE of the N treatment was significantly higher than that of non-N treatment under the $P_3$ treatment ($p < 0.05$). In 2021, the DMY, SD and EE of alfalfa under the N treatment were significantly higher than the non-N treatment

($p < 0.05$). The S/L was significantly lower than without N treatment ($p < 0.05$). The DMY, PH, SD, CP and EE of alfalfa were increased by 2.10~11.82%, 3.39~6.67%, 3.52~10.17%, 4.71~7.59% and 1.07~10.25%, and the S/L, NDF and ADF of alfalfa were decreased by 0.41~6.05%, 4.61~7.11% and 4.28~6.83% in the N application treatment compared with the non-N application treatment.

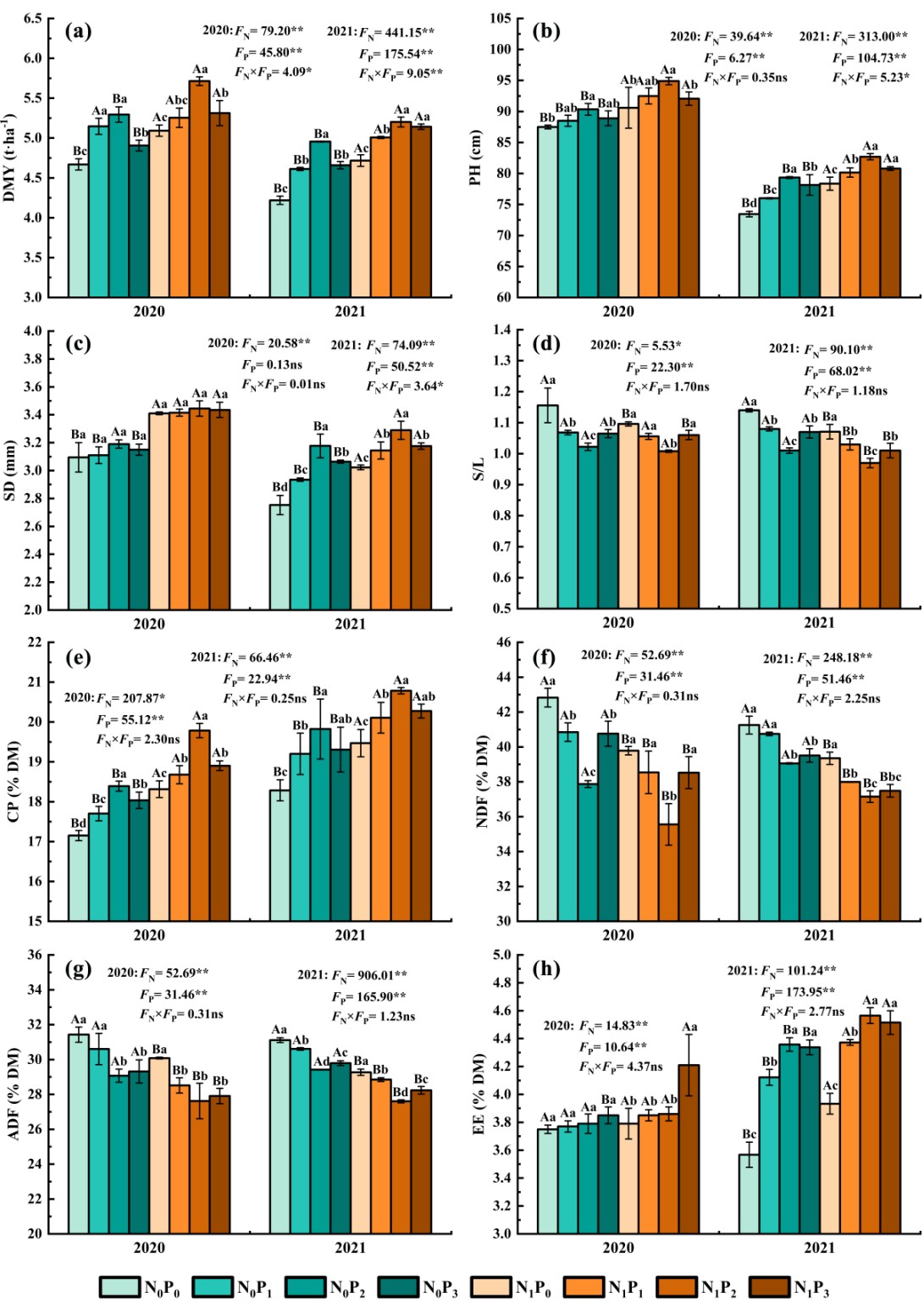

**Figure 1.** Dry matter yield (**a**), plant height (**b**), stem diameter (**c**), stem to leaf ratio (**d**), crude protein content (**e**), neutral detergent fiber (**f**), acid detergent fiber (**g**) and ether extract (**h**) of alfalfa under

different fertilization treatments in 2020 and 2021. All the values were the means of three replicates with standard errors. DMY, Dry matter yield; PH, Plant height; SD, Stem diameter; S/L, Stem to leaf ratio; CP, Crude protein content; NDF, Neutral detergent fiber; ADF, Acid detergent fiber; EE, Ether extract. Different capital letters indicate significant differences between different N fertilizer levels under the same P application condition ($p < 0.05$). Different small letters indicate significant differences between different P fertilizer treatments under the same N application condition ($p < 0.05$). $F_N$, $F_P$ and $F_N \times F_P$ represent the $F$ value under the N application levels, P application levels and the interaction of N and P application levels, respectively. ns indicates no significant difference ($p > 0.05$), * indicates significant difference ($p < 0.05$) and ** indicates extremely significant difference ($p < 0.01$).

### 3.2. Photosynthetic Performance

Under the same N application conditions, the $P_n$, $T_r$, $G_s$, WUE and $L_s$ of alfalfa leaves increased firstly and then decreased with increasing P application. In comparison, the $C_i$ of leaves dropped first and subsequently rose as the amount of P applied increased (Figure 2). $P_2$ treatment significantly increased the $P_n$, $T_r$, $G_s$ and $L_s$ of alfalfa leaves compared to non-P treatment ($p < 0.05$), and non-P treatment significantly increased the $C_i$ of alfalfa leaves compared to the other three P treatments ($p < 0.05$). There was no significant difference in WUE between different P application treatments in 2020 ($p > 0.05$). However, $P_2$ treatment was significantly greater than non-P treatment under the non-N treatment in 2021 ($p < 0.05$). The $P_n$, $T_r$, $G_s$, WUE and $L_s$ of alfalfa leaves were increased by 6.15~24.95%, 4.60~19.13%, 8.47~42.40%, 0.49~6.43% and 8.26~25.87%, and the $C_i$ of alfalfa leaves were decreased by 4.50~10.90% in the P application treatment compared with the non-P application treatment. Under the same P application circumstances, the $P_n$, $T_r$ and $L_s$ of alfalfa leaves treated with N were considerably greater than non-N treatment ($p < 0.05$), whereas the $C_i$ of leaves was significantly lower than that in N application treatment ($p < 0.05$). In 2020, the $G_s$ and WUE of alfalfa leaves treated with various N concentrations were not significantly different ($p > 0.05$), while the $G_s$ of alfalfa leaves under N treatment were significantly greater than those under non-N treatment in 2021 ($p < 0.05$). The $P_n$, $T_r$, $G_s$, WUE and $L_s$ of alfalfa leaves were increased by 4.95~11.93%, 4.24~11.02%, 0.15~20.41%, −0.75~2.28% and 2.72~14.89%, and the $C_i$ of alfalfa leaves were decreased by 2.72~14.89% in the N application treatment compared with the non-N application treatment.

The C3 light response model was applied to fit the $P_n$ to the PAR of alfalfa leaves under different treatments (Table 1, Figures 3 and 4). Under the same N application conditions, the $P_{max}$, AQY, $R_d$, LCP and LSP of alfalfa leaves increased first and then decreased with the increase in P application. The $P_{max}$, AQY, $R_d$ and LCP of alfalfa leaves treated with $P_2$ were significantly higher than the other three treatments ($p < 0.05$). Under the same P application conditions, the $P_{max}$, AQY, $R_d$, LCP and LSP of alfalfa leaves treated with N were higher than those without N treatment. The $R_d$, LCP and AQY of alfalfa were significantly higher than those without N treatment ($p < 0.05$). Under $P_1$ conditions, the $P_{max}$ of alfalfa leaves under N treatment was significantly higher than that without N treatment ($p < 0.05$). Under the conditions of $P_2$ and $P_3$, the AQY and $P_{max}$ of alfalfa leaves treated with N were significantly higher than those without N treatment ($p < 0.05$). In addition, there was no significant difference in the LSP of alfalfa leaves among the treatments under different N and P application conditions ($p > 0.05$).

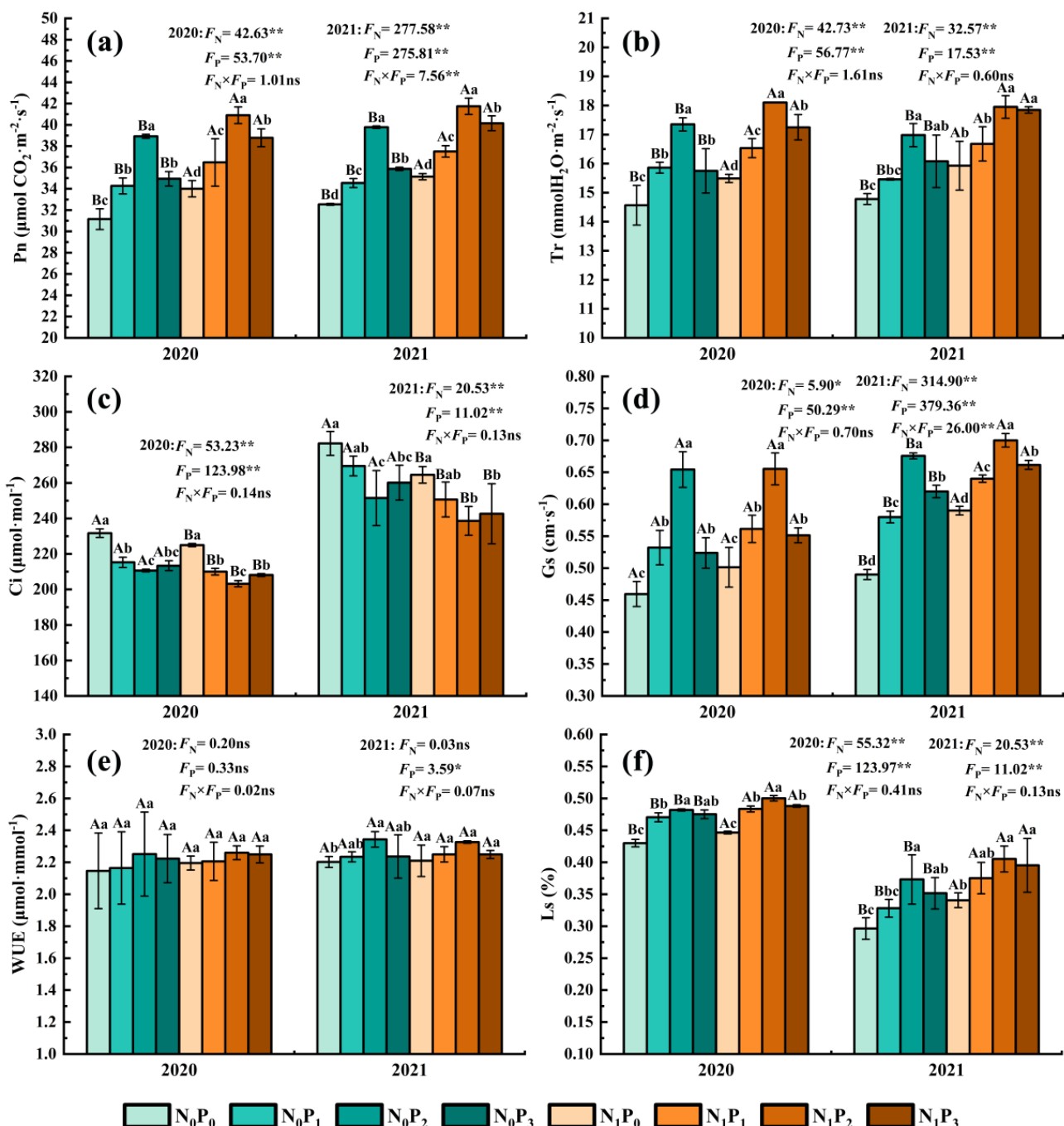

**Figure 2.** Net photosynthetic rate (**a**), intercellular $CO_2$ concentration (**b**), stomatal conductance (**c**), transpiration rate (**d**), instantaneous water use efficiency (**e**) and stomatal limit value (**f**) of alfalfa under different fertilization treatments in 2020 and 2021. All the values were the means of three replicates with standard errors. $P_n$, Net photosynthetic rate; $C_i$, Intercellular $CO_2$ concentration; $G_s$, Stomatal conductance; $T_r$, Transpiration rate; WUE, Instantaneous water use efficiency; $L_s$, Stomatal limit value. Different capital letters indicate significant differences between different N fertilizer levels under the same P application condition ($p < 0.05$). Different small letters indicate significant differences between different P fertilizer treatments under the same N application condition ($p < 0.05$). $F_N$, $F_P$ and $F_N \times F_P$ represent the F value under the N application levels, P application levels and the interaction of N and P application levels, respectively. ns indicates no significant difference ($p > 0.05$), * indicates significant difference ($p < 0.05$) and ** indicates extremely significant difference ($p < 0.01$).

**Table 1.** Light response characteristic parameters of alfalfa under different fertilization treatments.

| Treatments | $P_{max}$ [a]/(μmol $CO_2 \cdot m^{-2} \cdot s^{-1}$) | AQY/($CO_2 \cdot$Photon$^{-1}$) | $R_d$/(μmol $CO_2 \cdot m^{-2} \cdot s^{-1}$) | LCP/(μmol$\cdot m^{-2} \cdot s^{-1}$) | LSP/(μmol$\cdot m^{-2} \cdot s^{-1}$) |
|---|---|---|---|---|---|
| $N_0P_0$ | 32.4 ± 0.42 ad [b] | 0.0622 ± 0.001 bd | 2.77 ± 0.1 bc | 44.59 ± 2.58 bc | 2309.39 ± 24.39 aa |
| $N_0P_1$ | 35.34 ± 1.33 bc | 0.0658 ± 0.0008 ac | 3.35 ± 0.13 ab | 50.86 ± 2.28 ab | 2326.99 ± 14.01 aa |
| $N_0P_2$ | 41.91 ± 0.06 ba | 0.0691 ± 0.0001 ba | 4.06 ± 0.29 aa | 58.8 ± 2.06 aa | 2371.49 ± 9.51 aa |
| $N_0P_3$ | 38.63 ± 0.61 bb | 0.0674 ± 0.0008 bb | 3.44 ± 0.18 ab | 50.97 ± 3.04 ab | 2340.16 ± 10.84 aa |
| $N_1P_0$ | 33.36 ± 0.42 ad | 0.0666 ± 0.0012 ac | 3.34 ± 0.27 ac | 50.09 ± 2.33 ab | 2360.06 ± 8.94 aa |
| $N_1P_1$ | 37.3 ± 0.28 ac | 0.067 ± 0.0005 ac | 3.43 ± 0.18 abc | 51.24 ± 1.77 ab | 2382.16 ± 11.16 aa |
| $N_1P_2$ | 43.62 ± 1.23 aa | 0.0716 ± 0.0011 aa | 4.3 ± 0.21 aa | 60.09 ± 1.92 aa | 2402.6 ± 48.4 aa |
| $N_1P_3$ | 39.98 ± 0.67 ab | 0.0691 ± 0.0007 ab | 3.65 ± 0.16 ab | 52.88 ± 2.14 ab | 2384.16 ± 80.16 aa |
| $F_B$ [c] | 3.91 *[d] | 3.18 ns | 5.28 * | 6.95 ** | 0.05 ns |
| $F_N$ | 32.10 ** | 64.69 ** | 17.60 ** | 20.47 ** | 8.68 ** |
| $F_P$ | 255.71 ** | 70.01 ** | 51.42 ** | 102.36 ** | 1.99 ns |
| $F_N \times F_P$ | 0.67 ns | 5.37 * | 2.35 ns | 4.99 ** | 0.12 ns |

[a] $P_{max}$, Maximum net photosynthetic rate; LCP, Light compensation point; LSP, Light saturation point; AQY, Apparent quantum yield; $R_d$, Dark respiration rate. [b] Different capital letters indicate significant differences between different N fertilizer levels under the same P application condition ($p < 0.05$). Different small letters indicate significant differences between different P fertilizer treatments under the same N application condition ($p < 0.05$). [c] $F_N$, $F_P$ and $F_N \times F_P$ represent the $F$ value under the N application levels, P application levels and the interaction of N and P application levels, respectively. [d] ns indicates no significant difference ($p > 0.05$), * indicates significant difference ($p < 0.05$) and ** indicates extremely significant difference ($p < 0.01$).

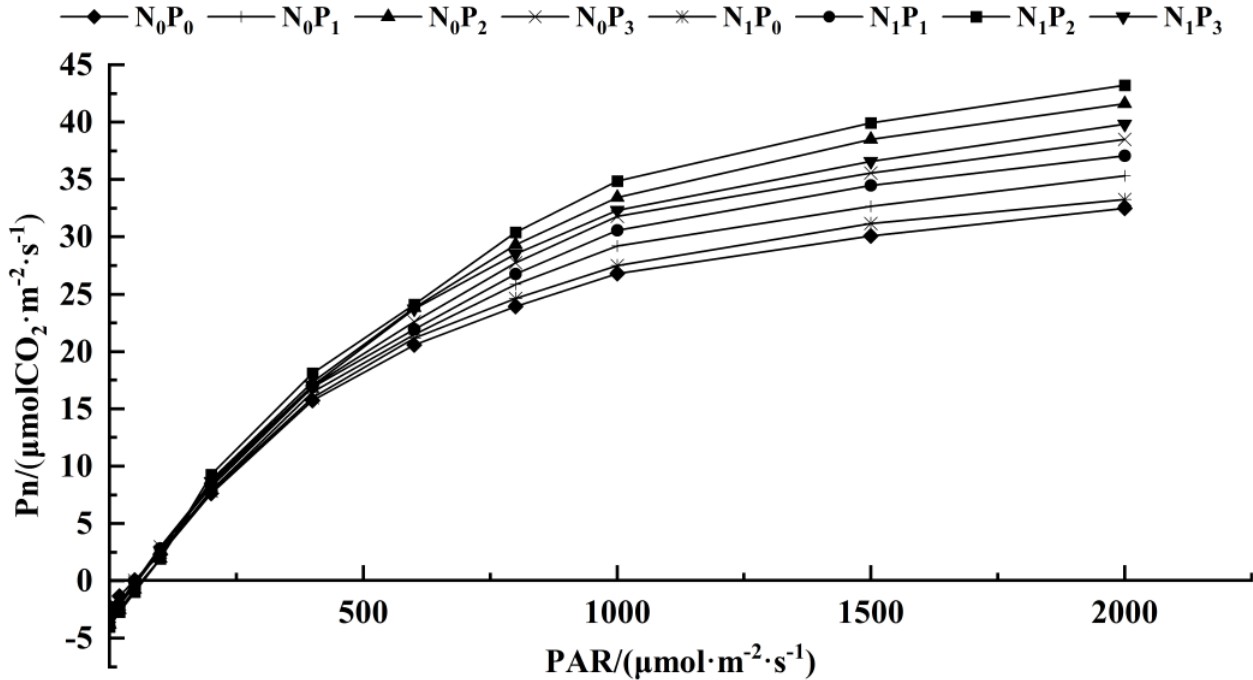

**Figure 3.** Photosynthesis–light intensity response curve of alfalfa leaves under different fertilization treatments.

The N application level had highly significant effects on the $P_{max}$, AQY, $R_d$, LCP and LSP of alfalfa leaves at the first flowering stage ($p < 0.01$); the N application level had an extremely significant effect on the AQY and the impact was the greatest, followed by the $P_{max}$, and the LSP was the lowest. The P application level had a highly significant effect on the $P_{max}$, AQY, $R_d$ and LCP of alfalfa leaves at the first flowering stage ($p < 0.01$), where the P application level had the greatest effect on the $P_{max}$, followed by LCP, AQY, $R_d$ and LSP, respectively. The interaction effect of N and P addition significantly ($p < 0.05$) or highly significantly ($p < 0.01$) affected the AQY and LCP, with the greatest effect on the AQY and the second highest on the LCP.

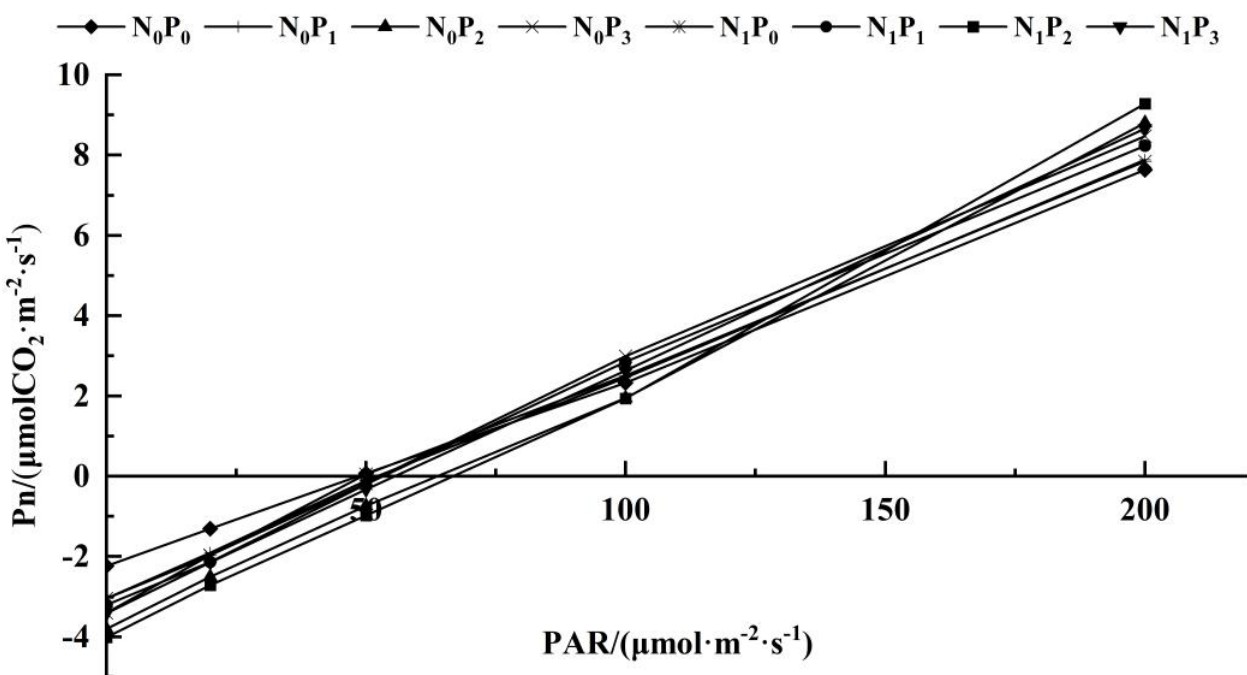

**Figure 4.** Response curve of photosynthesis to low light intensity in alfalfa leaves under different fertilization treatments. (PAR<200 $\mu$mol·m$^{-2}$·s$^{-1}$).

Under the same N application conditions, the Chl a, Chl b, Chl, Car, LA and SLW of alfalfa leaves showed a trend of first increasing and then decreasing with the increase in P application (Figure 5). The Chl a, Chl b, Chl, Car, LA and SLW of the $P_2$ treatment were significantly greater than those of non-P application treatment ($p < 0.05$). The Chl a, Chl b, Chl, Car, LA and SLW of alfalfa leaves were increased by 1.39~10.01%, 3.73~11.85%, 2.59~9.45%, 3.86~20.88%, 3.95~18.94% and 2.16~11.90% in the P application treatment compared with the non-P application treatment. Under the same conditions of P application, the LA of all N application treatments was significantly greater than non-N application treatments ($p < 0.05$), and in 2020, the Chl a and Chl of N application treatments were significantly greater than non-N application treatments under conditions of non-P application ($p < 0.05$), and the leaf Chl b and Chl of N application treatments were significantly greater ($p < 0.05$) than non-N application treatments under conditions of $P_2$, while the SLW in the N application treatment was significantly greater ($p < 0.05$) than in the non-N treatment under the conditions of $P_3$. In 2021, Chl a, Chl b, Chl, Car and SLW in the N application treatment were significantly greater ($p < 0.05$) than in the non-N treatment in 2021. The Chl a, Chl b, Chl, Car, LA and SLW of alfalfa leaves were increased by $-0.28$~4.61%, 1.68~5.01%, 0.30~4.62%, 0.61~7.11%, 5.15~9.26% and 2.02~7.12% in the N application treatment compared with the non-N application treatment.

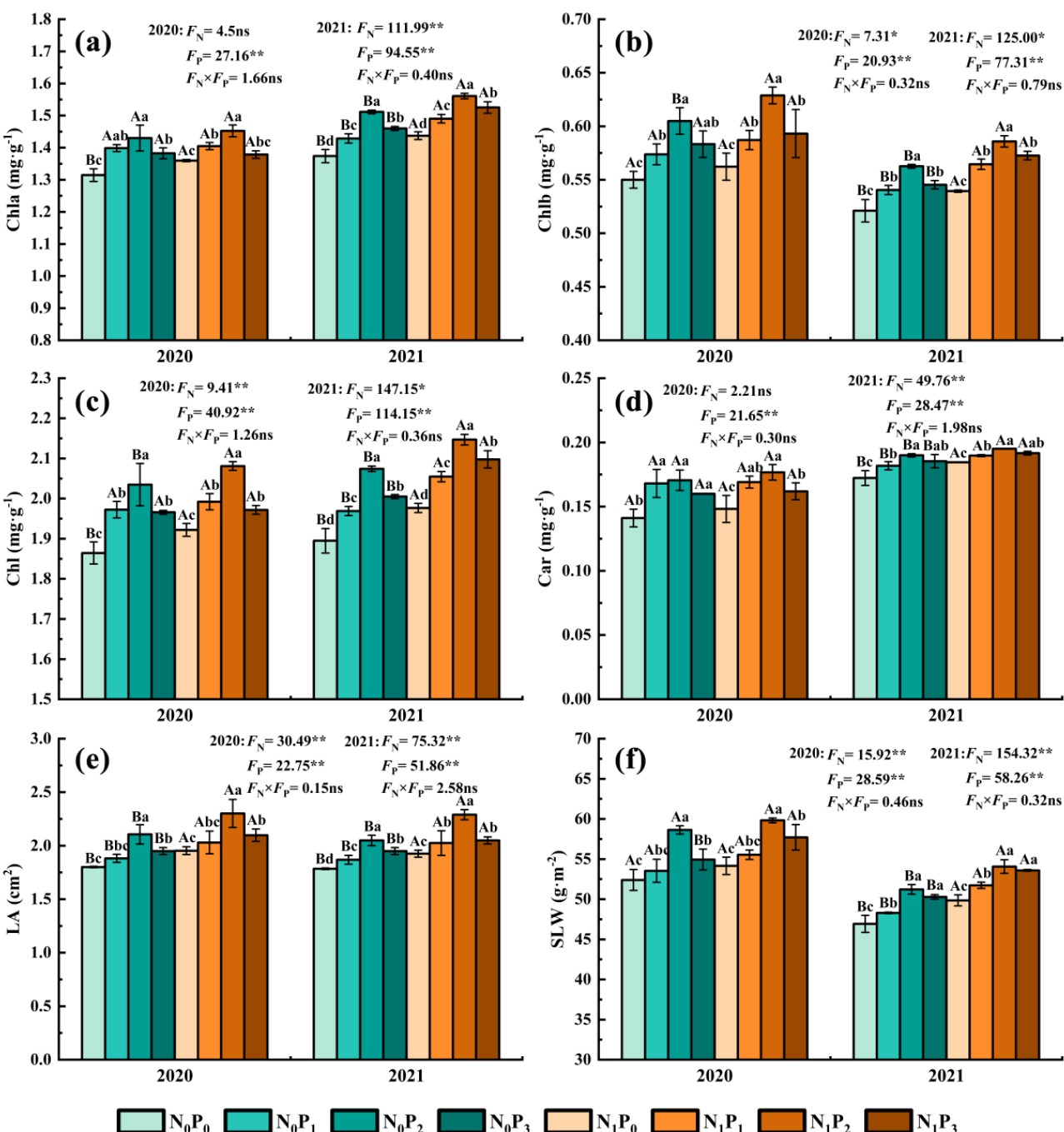

**Figure 5.** Chlorophyll a content (**a**), chlorophyll b content (**b**), sum of chlorophyll a and chlorophyll b (**c**), carotenoid content (**d**), leaf area (**e**) and specific leaf weight (**f**) of alfalfa under different fertilization treatments in 2020 and 2021. All the values were the means of three replicates with standard errors. Chla, Chlorophyll a content; Chlb, Chlorophyll b content; Chl, Sum of Chlorophyll a and Chlorophyll b; Car, Carotenoid content; LA, leaf area; SLW, Specific leaf weight. Different capital letters indicate significant differences between different N fertilizer levels under the same P application condition ($p < 0.05$). Different small letters indicate significant differences between different P fertilizer treatments under the same N application condition ($p < 0.05$). $F_N$, $F_P$ and $F_N \times F_P$ represent the $F$ value under the N application levels, P application levels and the interaction of N and P application levels, respectively. ns indicates no significant difference ($p > 0.05$), * indicates significant difference ($p < 0.05$) and ** indicates extremely significant difference ($p < 0.01$).

### 3.3. Anatomical Structure

Under the same N application conditions, the LT, PT, ST, CTR, SR and PT/ST of alfalfa showed a trend of increasing and then decreasing with the increase in P application (Table 2, Figure 6). All P$_2$ treatments were significantly greater than other treatments ($p < 0.05$), and the LT, UE, DE, PT, ST, CTR, SR and PT/ST of alfalfa were all greater than those of the treatments without N application. At the same time, the LT, PT and ST of alfalfa under the N application treatments were significantly greater ($p < 0.05$) than those without the N application.

**Table 2.** Anatomic structure indexes of alfalfa under different fertilization treatments.

| Treatments | LT [a]/(μm) | UE/(μm) | DE/(μm) | PT/(μm) | ST/(μm) | CTR/(%) | SR/(%) | PT/ST |
|---|---|---|---|---|---|---|---|---|
| N$_0$P$_0$ | 169.86 ± 5.8 bd [b] | 17.36 ± 1.15 aa | 18.67 ± 0.52 aa | 82.1 ± 2.1 bd | 51.52 ± 2.53 Bd | 48.34 ± 0.63 Ac | 30.37 ± 2.25 Aa | 1.6 ± 0.1 Ab |
| N$_0$P$_1$ | 178.15 ± 7.02 bc | 17.51 ± 1.36 aa | 18.67 ± 1.26 aa | 93.68 ± 4.5 bc | 54.83 ± 1.98 bc | 52.58 ± 0.67 ab | 30.84 ± 2.32 aa | 1.71 ± 0.14 aab |
| N$_0$P$_2$ | 205.66 ± 2.69 ba | 17.62 ± 1.22 aa | 19.51 ± 0.13 aa | 114.75 ± 0.81 ba | 63.82 ± 3.03 ba | 55.81 ± 1.04 aa | 31.02 ± 1.17 aa | 1.8 ± 0.09 aa |
| N$_0$P$_3$ | 197.41 ± 1.66 bb | 17.53 ± 0.44 aa | 18.99 ± 0.75 aa | 102.21 ± 4.97 bb | 60.5 ± 1.11 bb | 51.76 ± 2.11 bb | 30.65 ± 0.82 aa | 1.69 ± 0.11 aab |
| N$_1$P$_0$ | 179.98 ± 5.98 ad | 17.5 ± 0.18 aa | 18.75 ± 0.98 aa | 87.32 ± 2.95 ad | 54.91 ± 0.04 ad | 48.52 ± 0.02 ac | 30.53 ± 1aa | 1.59 ± 0.05 ab |
| N$_1$P$_1$ | 191.24 ± 2.97 ac | 17.68 ± 0.11 aa | 18.83 ± 0.85 aa | 102.62 ± 2.36 ac | 59.03 ± 2.28 ac | 53.68 ± 2.07 ab | 30.88 ± 1.67 aa | 1.74 ± 0.03 aab |
| N$_1$P$_2$ | 219.25 ± 0.9 aa | 17.79 ± 0.31 aa | 19.53 ± 0.99 aa | 123.48 ± 2.19 aa | 68.13 ± 0.65 aa | 56.32 ± 0.77 aa | 31.08 ± 0.17 aa | 1.81 ± 0.01 aa |
| N$_1$P$_3$ | 209.13 ± 2 ab | 17.52 ± 1.29 aa | 19 ± 0.54 aa | 112.65 ± 3.29 ab | 64.36 ± 2.12 ab | 53.86 ± 1.06 ab | 30.77 ± 0.72 aa | 1.75 ± 0.01 aab |
| $F_B$ [c] | 0.62 ns [d] | 2.40 ns | 4.91 ns | 3.32 ns | 1.28 ns | 2.38 ns | 0.14 ns | 0.08 ns |
| $F_N$ | 47.20 ** | 0.12 ns | 0.09 ns | 53.85 ** | 24.98 ** | 4.28 ns | 0.02 ns | 0.41 ns |
| $F_P$ | 93.80 ** | 0.11 ns | 1.90 ns | 164.34 ** | 51.78 ** | 44.98 ** | 0.16 ns | 6.05 ns |
| $F_N \times F_P$ | 0.19 ns | 0.02 ns | 0.03 ns | 0.95 ns | 0.70 ns | 0.81 ns | 0.01 ns | 0.16 ns |

[a] LT, Leaf total thickness; UE, Up-epidermal thickness; DE, Down-epidermal thickness; PT, Palisade tissue thickness; ST, Spongy tissue thickness; CTR, Compaction of leaf tissue; SR, Porosity of leaf tissue; PT/ST, Ratio between palisade tissue and spongy tissues. [b] Different capital letters indicate significant differences between different N fertilizer levels under the same P application condition ($p < 0.05$). Different small letters indicate significant differences between different P fertilizer treatments under the same N application condition ($p < 0.05$). [c] $F_N$, $F_P$ and $F_N \times F_P$ represent the $F$ value under the N application levels, P application levels and the interaction of N and P application levels, respectively. [d] ns indicates no significant difference ($p > 0.05$) and ** indicates extremely significant difference ($p < 0.01$).

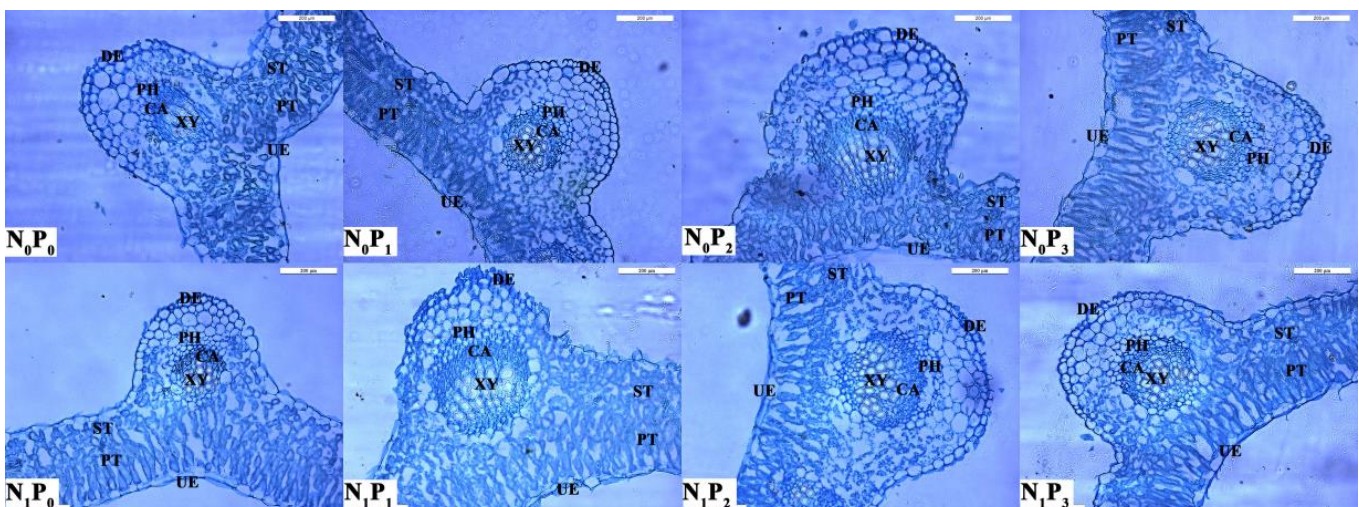

**Figure 6.** Anatomical changes in alfalfa leaves under different fertilization treatments. UE, Up-epidermis; PT, Palisade tissue; ST, Spongy tissue; DE, Down-epidermis; PH, Phloem; CA, Cambial; XY, Xylem; Scale bar = 200 μm.

The N application level had a highly significant effect on the LT, PT and ST of alfalfa at the first flowering stage ($p < 0.01$), with the greatest effect on PT, followed by LT, and the lowest on SR. The P application level had a highly significant effect on the LT, PT, ST and SR of alfalfa at the first flowering stage ($p < 0.01$), with the greatest effect on PT, the second greatest effect on LT, and the lowest effect on UE. The interaction effect of N and P had the greatest effect on PT, the second greatest effect on leaf CTR and the lowest effect on SR in alfalfa, and none of the effects on anatomical changes were significant.

### 3.4. Principal Component Analysis and Comprehensive Evaluation

The principal component analysis and comprehensive evaluation of the agronomic traits, photosynthetic performance and anatomical structure of alfalfa with the addition N and P are shown in Figure 7. Principal component analysis extracted the principal components with characteristic roots greater than 1 and a cumulative contribution rate greater than 85%, which can reflect most of the original data information. The agronomic traits (Figure 7a,e) extracted two principal components with characteristic roots greater than 1 and a cumulative contribution rate of 89.8%, and the comprehensive scores were $N_1P_2$(1.18) > $N_1P_3$(0.83) > $N_1P_1$(0.38) > $N_0P_2$(−0.02) > $N_1P_0$(−0.11) > $N_0P_3$(−0.45) > $N_0P_1$(−0.62) > $N_0P_0$(−1.20), from the highest to the lowest. Photosynthetic performance (Figure 7b,f) extracted two principal components with characteristic roots greater than 1 and a cumulative contribution rate of 95.1%, and the comprehensive scores were $N_1P_2$(1.33) > $N_0P_2$(0.75) > $N_1P_3$(0.33) > $N_1P_1$(0.22) > $N_0P_3$(−0.16) > $N_0P_1$(−0.24) > $N_1P_0$(−0.76) > $N_0P_0$(−1.48), from the highest to the lowest. The two principal components of anatomical structure (Figure 7c,g) were extracted with two characteristic roots greater than 1 and the cumulative contribution rate reached 92.1%, and the comprehensive scores were $N_1P_2$(1.29) > $N_0P_2$(0.57) > $N_1P_3$(0.41) > $N_1P_1$(0.22) > $N_0P_3$(0.04) > $N_0P_1$(−0.60) > $N_1P_0$(−0.61) > $N_0P_0$(−1.31). The principal component analysis (Figure 7d,h) of production performance, photosynthetic performance and anatomy extracted three principal components with a root greater than 1 and a cumulative contribution of 94.0%, and the comprehensive scores from high to low were $N_1P_2$(6.33) > $N_0P_2$(2.61) > $N_1P_3$(2.28) > $N_1P_1$(1.12) > $N_0P_3$(−0.93) > $N_0P_1$(−2.28) > $N_1P_0$(−2.70) > $N_0P_0$(−6.42).

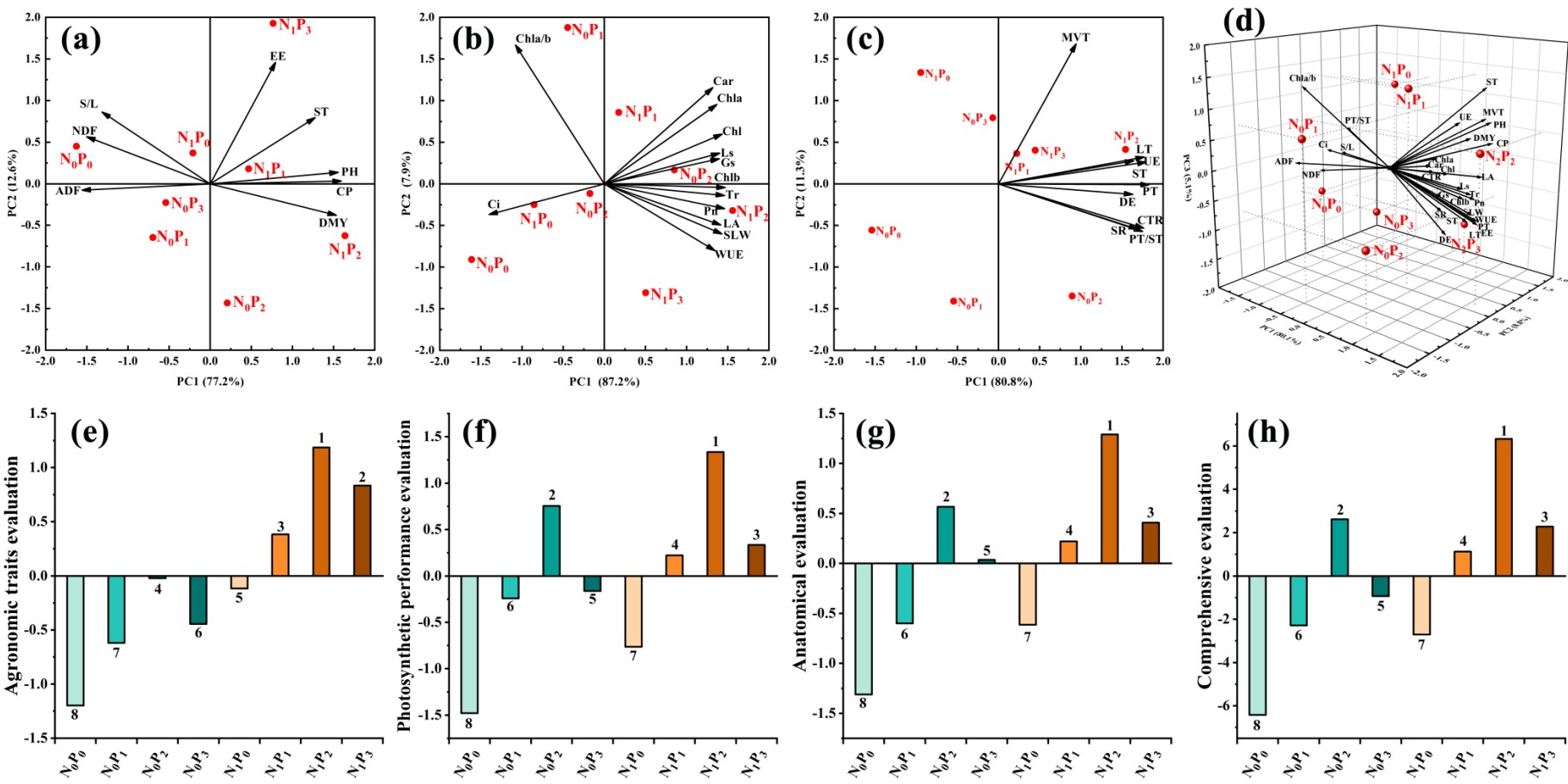

**Figure 7.** Principal component analysis (**a–d**) and comprehensive evaluation (**e–h**) of agronomic traits, photosynthetic properties and anatomical structure of alfalfa under N and P interaction.

### 3.5. Structural Equation Modeling Statistics

The structural equation model (Figures 8 and 9) showed the direct and indirect effects of key yield-influencing factors on DMY, which explained 77% of the DMY. Generally, $P_n$ ($r$d = 0.351, $r$d: direct regression weight coefficient) and $T_r$ ($r$d = 0.519) had a direct effect on DMY. ST directly ($r$d = 0.227) and indirectly ($r$i = 0.522, $r$i: indirect regression weight coefficient) influenced the formation of DMY through $P_n$ ($r$ = 0.557) and $T_r$ ($r$ = 0.630). PT directly ($r$d = −0.274) and indirectly ($r$i = 0.522) through $P_n$ ($r$ = 0.218), $T_r$ ($r$ = 0.213) and ST ($r$ = 0.807) affected the formation of DMY. Total Chl directly ($r$d = 0.109) and indirectly ($r$i = 0.654) through fenestrations ($r$ = 0.635), ST ($r$ = 0.169) and $P_n$ ($r$ = 0.557) affected the formation of DMY.

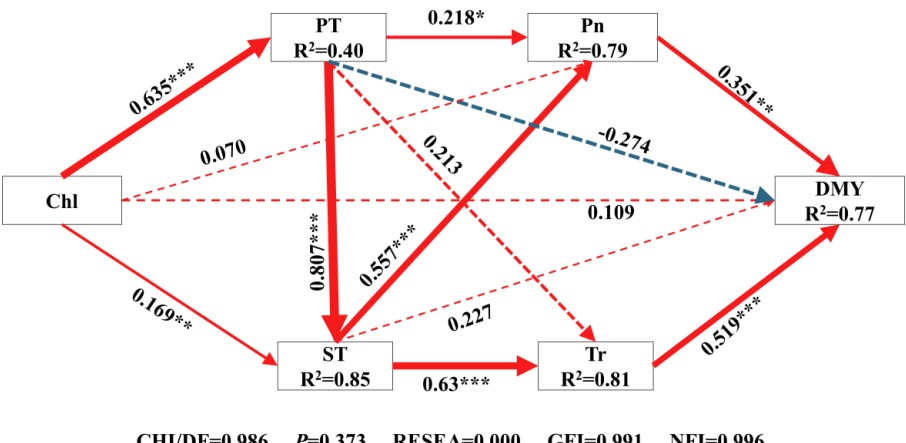

CHI/DF=0.986    *P*=0.373    RESEA=0.000    GFI=0.991    NFI=0.996

**Figure 8.** Comprehensive analysis of the relationship between five main influencing factors of yield and dry matter yield based on structural equation model. Chl, Chlorophyll content; PT, Palisade tissue thickness; ST, Spongy tissue thickness; $P_n$, Net photosynthetic rate; $T_r$, Transpiration rate; DMY, Dry matter yield. The one-way arrow in the figure indicates the causal relationship, the red solid line indicates a significant positive correlation, the blue dashed line indicates a positive correlation but not significant, and the green dashed line indicates a negative correlation but not significant. The numbers on the line are standardized path coefficients, and * indicates significance at the level of 0.05, ** means significance at the level of 0.01, *** means significance at the level of 0.001, and $R^2$ in the box of significant variables represents the total explanatory rate of all variables that point to the dependent variable.

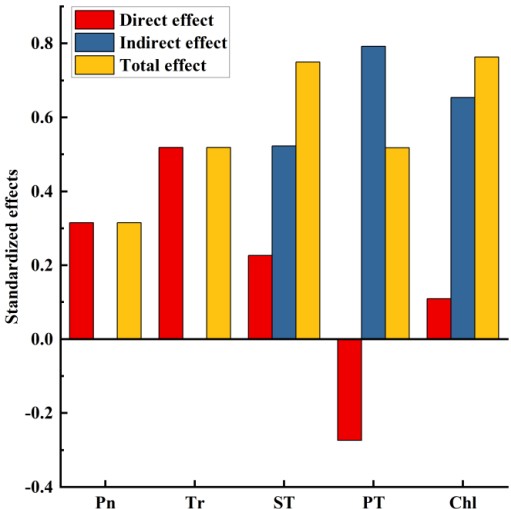

**Figure 9.** Normalized regression weights of direct, indirect and total effects of five major yield factors on dry matter yield. Chl, Chlorophyll content; PT, Palisade tissue thickness; ST, Spongy tissue thickness; $P_n$, Net photosynthetic rate; $T_r$, Transpiration rate.

## 4. Discussion

### 4.1. Effects of Nitrogen and Phosphorus Addition on Agronomic Characteristics of Alfalfa

Optimizing fertilization measures helps to promote the absorption and utilization of nutrients by plants, increase yield, maintain soil nutrients and prevent a deficiency or excessive accumulation of soil nutrients [19]. Our study found that the growth traits such as DMY and CP of the N application were higher than those of the non-N treatment under the same P application conditions, indicating that N application can promote plant growth and development, increase the productivity of alfalfa and improve nutritional quality. Other studies have shown that N application can increase the intra-plant N content, improve the dry matter and N transport capacity of nutrient organs [11] and promote the redistribution of photosynthetic products and nutrient elements to the above-ground part of the plant, thus achieving coordinated growth of the above-ground and below-ground parts [20]. Under the same N application conditions, growth traits such as DMY, PH, SD and CP of alfalfa showed a trend of increasing and then decreasing with the increase in P application; the S/L, NDF and ADF of alfalfa all showed a trend of decreasing initially and subsequently increasing with increasing P application. This indicates that the amount of P in the soil was insufficient to maintain alfalfa growth, and that the growth of alfalfa and the amount of P applied were positively correlated at this time. However, there is a threshold value for P in alfalfa plants, and when the P concentration exceeds this threshold value, this can have a negative effect on alfalfa's mechanism due to the redundancy of P, when the increase in the amount of P applied does not further promote alfalfa growth, or even inhibits the growth of alfalfa. Studies have shown that the elements have antagonistic and synergistic effects on each other, and that the accumulation of effective soil P caused by P application inhibits the uptake and utilization of K, Fe, Cu and Zn by alfalfa [21]. Other studies have indicated that excess P in the plant strengthens the respiration of alfalfa and excessive consumption of photosynthetic products, resulting in the dysregulation of nutritional and reproductive growth; moreover, stem and leaf growth is inhibited, thus reducing the biomass of alfalfa [22]. Alfalfa itself can adapt to changes in P levels by altering the root morphology, when excessive P content in the soil will inhibit the growth of alfalfa roots, which will affect the absorption and utilization of other elements [23]. In addition, excess P will also cause soil acidification, salinization, hardening and other problems [24].

### 4.2. Effects of Nitrogen and Phosphorus Addition on Photosynthetic Performance of Alfalfa

N and P are indispensable nutrients for plants' normal growth and development. This study discovered that N application significantly increased the chlorophyll content, which in turn encouraged an increase in photosynthetic rate, and also facilitated an increase in $G_s$ to facilitate the entry and exit of $CO_2$ and water during photosynthesis, which was also consistent with early research on wheat [25]. N is involved in the composition of many essential compounds, including proteins, nucleic acids, photosynthetic enzymes, etc. [26]. The rational application of N facilitates the formation of photosynthetic organs, energy transfer and transport of photosynthetic products, all of which in turn affect the photosynthesis of plants, improve their productivity and facilitate carbon fixation. Additionally, N was involved in the synthesis of photosynthetic pigments in leaves, which are the material basis of photosynthesis and have the role of capturing and transmitting light energy [27]; its content was positively and significantly correlated with photosynthesis. Our research found that P application also increased photosynthetic pigment; the P application significantly improved the $P_n$ and promoted an increase in $G_s$. Simultaneously, P application also increased the chlorophyll content, most likely because of the synergistic effect of increased exogenous P on alfalfa N uptake [28]. Other studies are basically consistent with ours: P contributes significantly to plant growth and development, photosynthetic regulation and yield quality formation by increasing the efficiency of photosynthetic carboxylation, promoting the formation of various photosynthetic enzymes and improving the uptake of elements such as N [29]. In addition, we discovered that increasing N and P could also

promote an increase in leaf area, increase the plant's ability to capture light energy and thus promote photosynthetic product output [30].

$C_i$ is an indispensable parameter for analyzing stomatal and non-stomatal limitation in photosynthesis plants, acting as a $CO_2$ mediator in the photosynthetic process, and was found to be influenced by the atmospheric $CO_2$ concentration as a source, on the one hand, and leaf $G_s$ and photosynthetic consumption on the other hand [31]. Our research found that $P_n$ showed a contrasting pattern of variation compared to $C_i$, which was consistent with another plant study [32]. This was most likely because the fertilization treatments improved the photosynthetic activity of alfalfa leaves, which resulted in higher $P_n$, with little causality with regard to intercellular $CO_2$ changes. Another study demonstrated that $P_n$ was positively correlated with $C_i$, which was more frequently the case when $G_s$ was restricted during the plant's "midday depression of photosynthesis", indicating that the reduction in photosynthetic rate was caused by the decrease in $G_s$ and consequently $C_i$ [33]. Moreover, the reduction in the $P_n$ of plant leaves should be mainly attributed to stomatal factors rather than non-stomatal factors under this variation pattern. Occasionally, a more complicated scenario occurs in which the two are uncorrelated [34], which might be the result of a combination of $C_i$ and photosynthetic activity limitations.

Increased N and P fertilization had a specific effect on the AQY of alfalfa leaves, and the slope of the photosynthetic response curve was greater in the fertilization treatment than in the control treatment under low light conditions, indicating that N and P have a facilitating effect on the AQY of alfalfa leaves and can improve the efficiency of light energy capture by alfalfa leaves [35]. The $P_{max}$ and $R_d$ of alfalfa leaves in the fertilized treatment were significantly higher than those in the unfertilized treatment, indicating that the fertilized treatment had greater photosynthetic potential under strong light. Meanwhile, the rate at which plants consume photosynthetic products increases, leading to an increase in the rate of dark respiration. The LSP and LCP represent plants' capacity to adapt to light, plants with high LSP and low LCP having an increased ability to adapt to light, and conversely, plants with a reduced ability to adapt to light have low LSP and high LCP [36]. Both the LSP and LCP of alfalfa leaves in the fertilized treatments were significantly increased compared to non-fertilized treatments, and other studies have proven that the increase in LCP is likely due to the fact that both the synthesis and transport of photosynthetic products require plants to respire for energy, thus requiring a greater photosynthetic rate to balance this [37].

*4.3. Effects of Nitrogen and Phosphorus Addition on Anatomical Structure of Alfalfa*

Changes in the anatomical structure of plant leaves are closely related to their physiological functions and external environment, and under long-term external fertilization, the morphological structure of leaves will undergo certain physiological adaptations to maximize photosynthetic capacity maintenance [18]. The mesophyll was the primary photosynthesis part of the leaves, and the thickness of PT and ST, as well as the morphological changes in palisade cells, will inevitably affect the leaves' photosynthetic efficiency [38]. Accordingly, an examination of the changes in the anatomical structure of alfalfa leaves may provide an explanation as to how photosynthesis characteristics have changed. Compared to the no fertilizer treatment, N and P application significantly increased the LT, PT and ST of alfalfa, and this was consistent with previous research [39]. This was because N and P deficiency will lead to changes in plant morphological structure, resulting in a reduction in leaf area and lower leaf transpiration, while excessive N and P supply will lead to imbalances in plant growth and development, which in turn will influence changes in plant anatomy. Plant leaves with a thicker PT under fertilization conditions will contain more chlorophyll, which will increase the plant's ability to access carbon. Additionally, because the palisade tissues were arranged perpendicularly to the vascular tissues and leaf surface, while fully utilizing light energy, they reduced the transpiration area and prevented the excessive evaporation of water, thereby enhancing the photosynthetic efficiency [14]. It was likely that the thickening of the leaf ST under the fertilization conditions reflected an increase in the leaf's internal porosity, which promotes gas exchange inside the leaf and

can further increase its Ci. In addition, the spongy tissue scatters the sunlight received by the palisade tissue, which means that more light is available to the leaves, thereby providing the possibility of an increased photosynthetic rate [40]. Furthermore, the increase in PT/ST contributes to the greater utilization of light energy, and the increased thickness of the upper epidermis of the leaves facilitated increased plant stress resistance, while the increased thickness of the lower epidermis facilitated the entry and exit of water and gas.

*4.4. Relationship between Agronomic Characteristics, Photosynthetic Performance and Anatomical Structure of Alfalfa under Nitrogen and Phosphorus Addition*

Changes in plant anatomical structure, photosynthetic efficiency and agronomic traits are closely linked. Photosynthetic physiological and biochemical indicators and anatomical structure changes can promote the enhancement of photosynthesis, which further contributes to alfalfa's dry matter yield, growth traits and nutritional quality [41]. In general, N can promote an increase in chlorophyll in plant leaves; the content of chlorophyll was positively correlated with leaf thickness. Moreover, chlorophyll was the main pigment involved in photosynthesis activity and played an important role in the absorption, conversion and transmission of light, and the content of chlorophyll was related to the photosynthetic rate of plants [24,42]. In this study, N and P were found to have a strong synergistic effect on the growth traits and photosynthetic physiology of alfalfa and did not simply superimpose their effects, which was the main reason that P application also increased the chlorophyll content. Increased N and P will also increase the thickness of the DE and ST of alfalfa, which facilitated the entry and exit of gas and water, and further affected the photosynthetic indexes of alfalfa leaves, such as $G_s$ and $T_r$. Increasing the thickness of UE and PT of leaves will also enhance the leaves' ability to adapt to strong light and further increase $P_{max}$ and LSP [14,43].

The synergistic improvement in the leaf photosynthetic rate and the photosynthetic area was essential to increase the dry matter yield in alfalfa [44]. N and P can promote the improvement of the photosynthetic performance of plants, mainly from two perspectives: the direct effect increased the chlorophyll content in plant leaves, increasing $P_n$, and enhancing the photosynthetic enzyme activity can effectively increase the conversion of $CO_2$ in alfalfa leaves, while increasing $G_s$ can facilitate the transport of $CO_2$ and water and thereby improve the photosynthetic efficiency [45]. The indirect effect was to increase the effective photosynthetic area by increasing the number of leaves, expanding the leaf area and delaying leaf senescence. In addition, photosynthesis was the largest source of carbon in alfalfa, and the use of light energy in alfalfa does not depend entirely on the photosynthetic rate and leaf area size, but may also be closely related to the photosynthetic angle, spatial arrangement, shape and size of alfalfa leaves, etc. Whether N and P application will affect these factors requires further investigation.

## 5. Conclusions

This study demonstrated the positive effects of alfalfa physiological and biochemical indicators and anatomy on alfalfa photosynthesis. The synergistic improvement in the photosynthetic efficiency and the photosynthetic area of alfalfa fertilization treatment further contributed to the improvement of alfalfa production performance. We found that $T_r$ ($r = 0.519$) and $P_n$ ($r = 0.351$) had a greater direct impact on the formation of dry matter yield, while Chl ($r = 0.763$) and ST ($r = 0.749$) had the greatest comprehensive impact. The increased application of N and P fertilizers increased the AQY of alfalfa leaves; the $P_{max}$ and LSP of alfalfa leaves under N application treatment were significantly higher than those of the non-N treatment. Under the influence of long-term external fertilization, the morphological structure of the leaves undergoes certain physiological adaptations. For example, N and P increase the thickness of the ST of alfalfa, which will facilitate the entry and exit of gas and water, and will further affect the photosynthetic indices, such as stomatal conductance and transpiration rate, of alfalfa leaves. Increased PT thickness will also enhance the adaptability of plant leaves to strong light, and then increase the $P_{max}$

and LSP. These results indicated that the application of N and P fertilizers contributed to the adaptation of alfalfa's leaf anatomy and the improvement of the photosynthetic capacity, which was beneficial to the improvement of alfalfa DMY, growth traits and nutritional quality, with the most obvious effect at 120 kg·ha$^{-1}$ of N and 100 kg·ha$^{-1}$ of P$_2$O$_5$ application.

**Author Contributions:** Conceptualization, Y.S., X.W., C.M. and Q.Z.; methodology, Y.S.; software, Y.S.; validation, Y.S. and Q.Z.; formal analysis, Y.S.; investigation, Y.S.; resources, Q.Z.; data curation, Y.S.; writing—original draft preparation, Y.S.; writing—review and editing, X.W., C.M. and Q.Z.; visualization, Y.S.; supervision, Q.Z.; project administration, Q.Z.; funding acquisition, Q.Z. All authors have read and agreed to the published version of the manuscript.

**Funding:** This research was funded by the National Natural Science Foundation of China (Grant no. 32001400), the Fok Ying Tung Education Foundation of China (Grant no. 171099) and the Science and Technology Innovation Key Talent Project of Xinjiang Production and Construction Corps (2021CB034).

**Institutional Review Board Statement:** Not applicable.

**Informed Consent Statement:** Not applicable.

**Data Availability Statement:** Not applicable.

**Conflicts of Interest:** The authors declare no conflict of interest. The funders had no role in the design of the study; in the collection, analyses, or interpretation of data; in the writing of the manuscript, or in the decision to publish the results.

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
