# Peer review of "Effects of Nitrogen and Phosphorus Addition on Agronomic Characters, Photosynthetic Performance and Anatomical Structure of Alfalfa in Northern Xinjiang, China"

_agronomy, doi:10.3390/agronomy12071613_

Round 1
Author Response
Response to Reviewer 1 Comments
Point 1: Abstract: Ok and need more quantify data.
Response 1: We have added more quantify data of several key indexes to the abstract, including the dry matter yield, crude protein, and net photographic rate and specific leaf weight. The details are as follows:
The results showed that the dry matter yield (DMY), crude protein (CP), net photosynthetic rate (Pn) and specific leaf weight (SLW) were increased by 2.10%~11.82%, 4.95%~11.93%, 4.71%~7.59% and 2.02%~7.12% in the N application treatment compared with the non-N application treatment, while the DMY, CP, Pn and SLW were increased by 3.19%~17.46%, 1.99%~8.42%, 6.15%~24.95% and 2.16%~11.90% in the P application treatment compared with the non-P application treatment.
Point 2: Introduction and objectives written well, need updated literature mainly in relation to productivity.
Response 2: According to the reviewer's opinion, we have founded the latest relevant references to replace the over aged ones.
Point 3: Materials and methods are ok, need modification in following points (Experimental design and crop management, and Photosynthetic indexes)
Response 3: According to the reviewer's opinion, We have simplified and reordered the materials and methods, and supplemented the environmental factor determination methods. The details are as follows:
2.2. Experimental design and crop management
A two-factor randomized complete block design with three replicates was implemented in each treatment. The experimental treatments were the factorial combinations of two gradients of N fertilizer (as N equivalent; 0 (N0), 120 kg·ha-1 (N1)), and four gradients of P fertilizer (as P2O5 equivalent; 0 (P0), 50 (P1), 100 (P2) and 150 (P3) kg·ha-1), a total of 8 treatments were applied with N and P interaction. The N and P addition trials were conducted for two consecutive years, during which no rhizobia inoculation was carried out. The N fertilizer used in the experiment were urea (containing N 46%) and mono-ammonium phosphate (containing P2O5 11%). The P fertilizer was mono-ammonium phosphate (containing P2O5 52%), N0 and N1 treatments were fertilized by adding N fertilizer to maintain the same level of N under different P treatments, and fertilizer was applied through a drip irrigation system in the highly efficient water and fertilizer saving way of "fertilizer follows water". Fertilizer was applied by drip irrigation system with water at the branching stage (April 19, 2020 and April 18, 2021) and 3-5 days after mowing of the first three cuts (May 25, July 4, August 14, 2020, May 27, July 3, August 4, 2021).
The WL366HQ alfalfa seeds was sown manually in strips in 2019 at a seeding rate of 18.0 kg·ha-1, a row spacing of 20 cm and a sowing depth of 2 cm. Deep ploughing of the land before sowing provides a suitable environment for the deep penetration of alfalfa roots, and timely suppression after sowing is conducive to the close combination of seed and soil, and the moisture retention of the soil surface. The drip irrigation tape was buried shallowly to 10 cm below the soil layer with a spacing of 60 cm. The specifications of each plot were 4 m×6 m, and a 1 m protection row was set between the plots to prevent the infiltration of nutrients. In addition to fertilization, other management measures such as watering, weeding and insect control are carried out uniformly following local high-yielding alfalfa field production habits.
2.3.2. Photosynthetic indexes
Sunny and cloudless weather was selected for field photosynthesis measurements using an LI-6400 portable photosynthesizer (LI-COR Inc., Lincoln, NE, USA) from 11:00 a.m. to 13:00 a.m. In the first flowering of the second cut of alfalfa (June 26, 2020 and June 28, 2021). Mature leaves of the same leaf position that were healthy and disease-free and of uniform growth were randomly selected for the measurement, and each treatment was repeated six times and the average value was taken, and the measurement indexes included net photosynthetic rate (Pn), intercellular CO2 concentration (Ci), stomatal conductance (Gs) and transpiration rate (Tr), meanwhile, environmental factors such as photosynthetically active radiation (PAR), field CO2 concentration (Ca), relative air humidity (RH), atmospheric temperature (Ta), leaf surface saturated vapor pressure (Vpdl) were recorded. The calculation formulas for instantaneous water use efficiency (WUE) and stomatal limit value (Ls) are as follows:
(1)
(2)
Using the LI-6400 portable photosynthesis instrument and the LI-6400-02B red and blue light source to measure the light response parameters of the leaves. The leaves were induced under 1200 μmol·m-2·s-1 light intensity for 20 min before measurement. The photosynthetically active radiation (PAR) gradient was set to 2000, 1500, 1000, 800, 600, 400, 100, 50, 20 and 0 μmol·m-2·s-1. Maximum net photosynthetic rate (Pmax), light compensation point (LCP) and light saturation point (LSP) were calculated by fitting photosynthetic data using the C3 plant light response model [17], and apparent quantum yield (AQY) and dark respiration rate (Rd) were calculated using a response linear fit of the photosynthetic-low light intensity below 200 μmol·m-2·s-1.
Point 4: Results: Agronomic traits: Quantification is missing
Response 4: According to the reviewer's opinion, I has added more quantitative data to the results. The details are as follows:
3.1. Agronomic traits
Under the same N application conditions, the DMY, PH, SD and CP of alfalfa tended to increase first and then decrease with increasing P application. In contrast, S/L, NDF and ADF tended to drop initially and subsequently rise as P application increased (Figure 1). The PH, CP and DMY in P2 treatment were significantly greater than those in non-P treatment (P<0.05), but the NDF, ADF and S/L in non-P treatment were significantly lower (P<0.05). In 2020, the EE of the P3 treatment was significantly higher than the other three treatments (P<0.05), and in 2021, the EE of P2 and P3 treatments were significantly greater than those of P0 and P1 treatments (P<0.05). Furthermore, although there was no significant difference in SD among the P application treatments in 2020 (P>0.05), the SD for the P2 treatment will be significantly greater than the other three treatments in 2021 (P<0.05). The DMY, PH, SD, CP, EE of alfalfa were increased by 3.19%~17.46%, 1.14%~8.03%, 0.14%~15.4%, 1.99%~8.42%, 0.53%~22.16% and the S/L, NDF, ADF of alfalfa were decreased by 3.26%~11.55%, 1.24%~11.57%, 1.44%~8.19% in the P application treatment compared with the non-P application treatment. Under the identical P application conditions, N application resulted in significantly greater PH and CP than non-N treatment (P<0.05), but non-N treatment resulted in significantly lower NDF and ADF (P<0.05). In 2020, the DMY and SD of the N treatment were significantly greater than those of the non-N treatment under the P0 and P3 treatments (P<0.05), and the S/L of the N treatment was lower than that of the non-N treatment (P<0.05), and the EE of N treatment was significantly higher than that of non-N treatment under the P3 treatment (P<0.05). In 2021, the DMY, SD and EE of alfalfa under the N treatment will significantly be higher than the non-N treatment (P<0.05). The S/L was significantly lower than without N treatment (P<0.05). The DMY, PH, SD, CP, EE of alfalfa were increased by 2.10%~11.82%, 3.39%~6.67%, 3.52%~10.17%, 4.71%~7.59%, 1.07%~10.25% and the S/L, NDF, ADF of alfalfa were decreased by 0.41%~6.05%, 4.61%~7.11%, 4.28%~6.83% in the N application treatment compared with the non-N application treatment.
Point 5: Photosynthetic Performance: Quantification is missing
Response 5: According to the reviewer's opinion, we have added more quantitative data to the results. The details are as follows:
3.2. Photosynthetic Performance
Under the same N application conditions, the Pn, Tr, Gs, WUE and Ls of alfalfa leaves increased firstly and then decreased with increasing P application. In comparison, the Ci of leaves drops first and subsequently rises as the amount of P applied increases (Figure 2). P2 treatment significantly increased the Pn, Tr, Gs and Ls of alfalfa leaves compared to non-P treatment (P<0.05), and non-P treatment significantly increased the Ci of alfalfa leaves compared to the other three P treatments (P<0.05), there was no significant difference in WUE between different P application treatments in 2020 (P>0.05). However, P2 treatment was significantly greater than non-P treatment under the non-N treatment in 2021 (P<0.05). The Pn, Tr, Gs, WUE, Ls of alfalfa leaves were increased by 6.15%~24.95%, 4.60%~19.13%, 8.47%~42.40%, 0.49%~6.43%, 8.26%~25.87% and the Ci of alfalfa leaves were decreased by 4.50%~10.90% in the P application treatment compared with the non-P application treatment. Under the same P application circumstances, Pn, Tr and Ls of alfalfa leaves treated with N were considerably greater than non-N treatment (P<0.05), whereas Ci of leaves was significantly lower than N application treatment (P<0.05). In 2020, the Gs and WUE of alfalfa leaves treated with various N concentrations were not significantly different (P>0.05), while the Gs of alfalfa leaves under N treatment were significantly greater than those under non-N treatment in 2021 (P<0.05). The Pn, Tr, Gs, WUE, Ls of alfalfa leaves were increased by 4.95%~11.93%, 4.24%~11.02%, 0.15%~20.41%, -0.75%~2.28%, 2.72%~14.89% and the Ci of alfalfa leaves were decreased by 2.72%~14.89% in the N application treatment compared with the non-N application treatment.
Under the same N application conditions, the Chl a, Chl b, Chl, Car, LA and SLW of alfalfa leaves showed a trend of first increasing and then decreasing with the increase of P application (Figure 5). Chl a, Chl b, Chl, Car, LA and SLW of P2 treatment were significantly greater than those of non-P application treatment (P<0.05). The Chl a, Chl b, Chl, Car, LA, SLW of alfalfa leaves were increased by 1.39%~10.01%, 3.73%~11.85%, 2.59%~9.45%, 3.86%~20.88%, 3.95%~18.94%, 2.16%~11.90% in the P application treatment compared with the non-P application treatment. Under the same conditions of P application, the LA of all N application treatments was significantly greater than non-N application treatments (P<0.05), and in 2020, Chl a and Chl of N application treatments were significantly greater than non-N application treatments under conditions of non-P application (P<0.05), and leaf Chl b and Chl of N application treatments were significantly greater (P<0.05) than non-N application treatments under conditions of P2, and the SLW in the N application treatment was significantly greater (P<0.05) than in the non-N treatment under conditions of P3. In 2021, Chl a, Chl b, Chl, Car and SLW in the N application treatment were significantly greater (P<0.05) than in the non-N treatment in 2021. The Chl a, Chl b, Chl, Car, LA, SLW of alfalfa leaves were increased by -0.28%~4.61%, 1.68%~5.01%, 0.30%~4.62%, 0.61%~7.11%, 5.15%~9.26%, 2.02%~7.12% in the N application treat-ment compared with the non-N application treatment.
Point 6: Principal component analysis and comprehensive evaluation: Quantification is missing
Response 6: According to the reviewer's opinion, we have added more quantitative data to the results. The details are as follows:
The principal component analysis and comprehensive evaluation of agronomic traits, photosynthetic performance and anatomical structure of alfalfa under adding N and P were shown in Figure 7. Principal component analysis extracted the principal components with characteristic roots greater than 1 and the cumulative contribution rate greater than 85%, which can reflect most of the original data information. The agronomic traits (Figure 7a and e) extracted two principal components with characteristic roots greater than 1 and a cumulative contribution rate of 89.8%, and the comprehensive scores were N1P2(1.18)>N1P3(0.83)>N1P1(0.38)>N0P2(-0.02)>N1P0(-0.11)>N0P3(-0.45)>N0P1(-0.62)>N0P0(-1.20) from the highest to the lowest. Photosynthetic performance (Figure 7b and f) extracted two principal components with characteristic roots greater than 1 and a cumulative contribution rate of 95.1%, and the comprehensive scores were N1P2(1.33)>N0P2(0.75)>N1P3(0.33)>N1P1(0.22)>N0P3(-0.16)>N0P1(-0.24)>N1P0(-0.76)>N0P0(-1.48) from the highest to the lowest. The two principal components of anatomical structure (Figure 7c and g) were extracted with two characteristic roots greater than 1 and the cumulative contribution rate reached 92.1%, and the comprehensive scores were N1P2(1.29)>N0P2(0.57)>N1P3(0.41)>N1P1(0.22)>N0P3(0.04)>N0P1(-0.60)>N1P0(-0.61)>N0P0(-1.31). The principal component analysis (Figure 7d and h) of production performance, photosynthetic performance and anatomy extracted three principal components with a root greater than 1 and a cumulative contribution of 94.0%, and the comprehensive scores from high to bottom are N1P2(6.33)>N0P2(2.61)>N1P3(2.28)>N1P1(1.12)>N0P3(-0.93)>N0P1(-2.28)>N1P0(-2.70)>N0P0(-6.42).
Point 7: Discussion – The whole discussion needs to be updated literature.
Response 7: According to the reviewer's opinion, we have founded the latest relevant references to replace the over aged ones.
Point 8: Conclusion – Need quantification in relation to productivity and climate change.
Response 8: According to the reviewer's opinion, I have added more quantitative data to the conclusion. The details are as follows:
This study demonstrated the positive effects of alfalfa physiological and biochemical indicators and anatomy on alfalfa photosynthesis, the synergistic improvement in photosynthetic efficiency and the photosynthetic area of alfalfa fertilization treatment further contributed to the improvement of alfalfa production performance. We found that Tr(r=0.519) and Pn(r=0.351) had a greater direct impact on the formation of dry matter yield, while Chl(r=0.763) and ST(r=0.749) had the greatest comprehensive impact. The increased application of N and P fertilizers increased the AQY of alfalfa leaves, the Pmax and LSP of alfalfa leaves under N application treatment were significantly higher than those of the non-N treatment. Under the influence of long-term external fertilization, the morphological structure of the leaves undergoes certain physiological adaptations. For example, N and P increase the thickness of the ST of alfalfa, which will facilitate the entry and exit of gas and water, and will further affect photosynthetic indices, such as stomatal conductance and transpiration rate of alfalfa leaves. Increased PT thickness will also enhance the adaptability of plant leaves to strong light, and then increase the Pmax and LSP. These results indicated that the application of N and P fertilizers contributed to the adaptation of alfalfa leaf anatomy and the improvement of photosynthetic capacity, which was beneficial to the improvement of alfalfa DMY, growth traits and nutritional quality, with the most obvious effect at 120 kg·ha-1 of N and 100 kg·ha-1 of P2O5 application.

Reviewer 2 Report
Manuscript entitled “Effects of nitrogen and phosphorus addition on agronomic characters, photosynthetic performance and anatomical structure of alfalfa under drip irrigation” by Sun and Zhang.
Authors reported that productivity of alfalfa is associated with a large amount of NP requirements. Author tries to establish relationship between changes in alfalfa productivity and photosynthetic physiology and anatomy. They recommended that fertilization treatment showed the highest utilization efficiency for low light and better adaptation to strong light, but the Rd decreased.
They also suggested that the application of N and P fertilizers contributed to the adaptive changes of alfalfa leaf anatomy and the improvement of photosynthetic capacity, which were beneficial to the improvement of alfalfa dry matter yield, growth traits and nutritional quality, with the most obvious improvement effect at the application of 120 kg·ha-1 of N and 100 kg·ha-1 of P2O5.
Abstract: Ok and need more quantify data.
Introduction and objectives written well, need updated literature mainly in relation to productivity.
Materials and methods are ok, need modification in following points (Experimental design and crop management, and Photosynthetic indexes)
Results: Agronomic traits-: Quantification is missing
All Figure: Excellent
Photosynthetic Performance: Quantification is missing
Anatomical structure, Principal component analysis and comprehensive evaluation: Quantification is missing
Structural equation modeling statistics; Excellent
Discussion – The whole discussion needs to be updated literature.
Conclusion – Need quantification in relation to productivity and climate change.
Author Response
Response to Reviewer 2 Comments
Point 1: In the chapter "Materials and Methods", the test and evaluation methods used in the experimental work are described in detail. The description of the test methods is very precise and sufficiently detailed. The method of application of the drip irrigation tape and the exact time of application of the nutrient were also described. My question is how was the harvesting carried out after the application? How did you manage to check the clogging of the drip irrigation tape covered with soil?
Response 1: When the early flowering stage of Alfalfa comes, we use scissors and sickles to sample and measure yield. After that, we use portable mowers to cut the uncut areas.
There are two main aspects to observe whether the drip irrigation belt is clogged. On the one hand, the end of the drip irrigation belt is slightly beyond the experimental plot. During irrigation, we will observe whether the end soil is wet to judge whether there is damage or clog in the middle of the drip irrigation belt, but this rarely happens; On the other hand, we will check whether the soil surface in the middle of the drip irrigation zone is wet to judge whether it is blocked or not during the irrigation 3-5 days after each alfalfa cutting. In addition, at the beginning of the experiment, we also installed filters at the joints of drip irrigation belts and the water for irrigation is pumped from the pool by the water pump (to provide sufficient pressure for the drip irrigation belt), and the water pumped from the well to the pool will be precipitated to reduce the content of impurities in the water.
Point 2:The authors describe in detail the tested parameters and the methodology, Not all factors are defined in the stomatal limit value (Ls) formula (Ca?).
Response 2: We are sorry that we lack the description of environmental factors in the method of this article. Now it has been added, where Ca is the field atmospheric CO2 concentration.
Point 3: In the "Results" chapter, the tables are clear, the statistical analysis is thorough and the analysis of correlations is well justified. The figures are informative, but the quality is not good, please improve.
Response 3: The figures may not be very clear due to MS word compression. We have packaged the figures source file and sent it to the editor, we also adjusted the font size and scale of some drawings before.
Point 4: In Table 1 it is difficult to understand what 'A' refers to and what 'a' refers to. The relevant literature references are also provided in this chapter and in the "Conclusion" chapter as well, but are rarely compared to the author's own findings. Overall, the paper is well structured and well derived, and the statistical results are adequate. The study has shown positive effects of alfalfa physiological and biochemical indicators and anatomy on alfalfa photosynthesis, synergistic improvement of photosynthetic efficiency and improvement of alfalfa yield.
Response 4: I'm sorry that we didn't explain the significance analysis annotation clearly. Now it has been added to the table’s annotation. The details are as follows:
Different capital letters indicated significant difference between different N fertilizer levels under the same P application condition (P<0.05). Different small letters indicated significant difference between different P fertilizer treatments under the same N application condition (P<0.05).
We also revised some of the discussions and added more comparisons with other studies.

Reviewer 3 Report
Review report
The objective of the manuscript is to examine the application of exogenous N and P fertilizers on growth of alfalfa. The relationship between changes in alfalfa productivity and photosynthetic physiology and anatomy were also examined under drip irrigation. The manuscript has scientific potential and concluded that the application of P2O5 can improve the alfalfa dry matter yield, growth traits and nutritional quality.
Comments for authors:
L39: Use acryonym throughout the manuscript.
The title of the manuscript highlights drip irrigation, but no results were discussed in its relevance. Kindly improve.
Introduction: The introduction needs to shortened and more focused on title of the manuscript.
Materials and Methods: The parameters can be shortened, rather than full description of each trait analyzed.
Data analysis: Needs to rewritten. The statistical method of each trait can be discussed in materials and methods. In data analysis, discuss only the models used for each observation.
Discussion: More references can be added in discussion part to support the present study.
Conclusion: needs to be precise and highlights only the major findings of the study.

Author Response
Response to Reviewer 3 Comments
Point 1: L39: Use acryonym throughout the manuscript.
Response 1: According to the reviewers' opinions, we have abbreviated and supplemented AN and AP. The details are as follows:
Soil available nitrogen (AN) and available phosphorus (AP) deficiency are some of the most important factors limiting the productivity of alfalfa.
Point 2: The title of the manuscript highlights drip irrigation, but no results were discussed in its relevance. Kindly improve.
Response 2: We refer to the reviewers' opinions and decide to remove the drip irrigation in the title because it is not our experimental treatment. The details are as follows:
Effects of nitrogen and phosphorus addition on agronomic characters, photosynthetic performance and anatomical structure of alfalfa in Northern Xinjiang, China
Point 3: Introduction: The introduction needs to shortened and more focused on title of the manuscript.
Response 3: According to the reviewers' suggestions, we have partially deleted the introduction.
Point 4: Materials and Methods: The parameters can be shortened, rather than full description of each trait analyzed.
Response 4: According to the reviewers' opinions, we have partially simplified materials and methods. The details are as follows:
2.3. Sampling and measurements
2.3.1. Agronomic traits
Dry matter yield (DMY) was measured by the sampling method at the early flowering stage of alfalfa (flowering 5%~10%). Meanwhile, plant height (PH) stem diameter (SD) and Stem to leaf ratio (S/L) were measured.
Nutritional quality: Total nitrogen content was determined using the Kjeldahl method, and total crude protein (CP) was calculated by multiplying the obtained results with 6.25, the neutral detergent fiber (NDF) and acid detergent fiber (ADF) content were determined by the Van Soest method, and the ether extract (EE) was determined by the ether extraction method.
2.3.2. Photosynthetic indexes
Sunny and cloudless weather was selected for field photosynthesis measurements using an LI-6400 portable photosynthesizer (LI-COR Inc., Lincoln, NE, USA) from 11:00 a.m. to 13:00 a.m. In the first flowering of the second cut of alfalfa (June 26, 2020 and June 28, 2021). The measurement indexes included net photosynthetic rate (Pn), intercellular CO2 concentration (Ci), stomatal conductance (Gs) and transpiration rate (Tr), meanwhile, environmental factors such as photosynthetically active radiation (PAR), field CO2 concentration (Ca), relative air humidity (RH), atmospheric temperature (Ta), leaf surface saturated vapor pressure (Vpdl) were recorded. The calculation formulas for instantaneous water use efficiency (WUE) and stomatal limit value (Ls) are as follows:
(1)
(2)
Using the LI-6400 portable photosynthesis instrument and the LI-6400-02B red and blue light source to measure the light response parameters of the leaves. The leaves were induced under 1200 μmol·m-2·s-1 light intensity for 20 min before measurement. The photosynthetically active radiation (PAR) gradient was set to 2000, 1500, 1000, 800, 600, 400, 100, 50, 20 and 0 μmol·m-2·s-1. Maximum net photosynthetic rate (Pmax), light compensation point (LCP) and light saturation point (LSP) were calculated by fitting photosynthetic data using the C3 plant light response model [17], and apparent quantum yield (AQY) and dark respiration rate (Rd) were calculated using a response linear fit of the photosynthetic-low light intensity below 200 μmol·m-2·s-1.
2.3.3. Photosynthetic physiological and biochemical indexes
Photosynthetic pigment: Chlorophyll content (Chl) and carotenoid content (Car) were determined by the spectrophotometer method [18].
Leaf area: The leaves were brought back to the laboratory, scanned using a scanner to form images, and leaf area (LA) was calculated using Image Pro Plus software (Media Cybernetics, Silver Spring, MD, USA).
Specific leaf weight (SLW): The whole green mature leaves of alfalfa were cut, and their total area was determined by the above-mentioned leaf area determination method, then dried and weighed, repeated 3 times, and the average value was taken. Specific leaf weight is the ratio of the dry matter weight of whole alfalfa leaves to leaf area.
2.3.4. Anatomical structure
To analyze the anatomical structure of alfalfa leaves, before cutting the alfalfa, the mature and healthy fresh leaves at the same leaf position (middle leaflet of the trifoliate compound leaf in the 3rd leaf position) were taken before alfalfa mowing and immediately fixed in FFA, after immersing for 12 hours, paraffin sections were made Photographs were taken at 100× magnification using a computer connected to an Olympus BX43F microscope (Olympus Inc., Tokyo, Japan). Leaf total thickness (LT), Main vein thickness (MVT), up-epidermal thickness (UE), down-epidermal thickness (DE), Palisade tissue thickness (PT) and spongy tissue thickness (ST) were measured using the software tools provided in Image Pro Plus 6.1 and averaged over multiple measurements. The specific formulas for the compaction of leaf tissue (CTR) and porosity of leaf tissue (SR) are as follows:
(3)
(4)
Point 5: Data analysis: Needs to rewritten. The statistical method of each trait can be discussed in materials and methods. In data analysis, discuss only the models used for each observation.
Response 5: According to the reviewers' opinions, I have rewritten data analysis, deleted some unnecessary things, and described some in the materials and methods. The details are as follows:
2.4. Data analysis
The effects of N and P on the indicators of alfalfa were examined using a two-way (N, P, N×P) ANOVA for each year, and multiple comparisons were performed using Duncan's method. The assumptions of normality (Shapiro-Wilk test) and homogeneity (Bartlett’s test) were tested before subjecting the data to ANOVA. Structural equation modeling (SEM) was performed using IBM SPSS AMOS 24 (SPSS Inc., Chicago, IL, USA) software to predict the interactions among chlorophyll, palisade tissue, spongy tissue, Pn, Tr and DMY, and several model fitness parameters were used to judge the reasonableness of the model, mainly including the ratio of chi-square value to the degree of freedom (CHI/DF<3), test P-value (P>0.05), asymptotic residual mean square and root square (RMSEA<0.08), goodness of fit index (GFI>0.9), and value-added fit number (NFI>0.9), and when the model met these conditions, it indicated reasonable model fitness.
The C3 plant light response model is:
(5)
Among them, I is photosynthetically active radiation; α, β, γ and ε are 4 coefficients. Set the initial values of α, β, γ and ε to 0.01, 0.0001, 0.001 and 0.5, respectively, and substitute the light response curve data and initial values into the model equation to calculate the coefficients α, β, γ and ε of the model by the Myquart method.
The calculation formula of the Pmax is:
(6)
The calculation formula of the LCP is:
(7)
The calculation formula of the LSP is:
(8)
The linear fitting equation for a low light intensity-net photosynthetic rate below 200 μmol·m-2·s-1 is:
(9)
Where δ is the slope of the straight line fitting equation. When I=0 μmol·m-2·s-1, Pn is the Rd; when I=200 μmol·m-2·s-1, δ is the AQY.
Point 6: Discussion: More references can be added in discussion part to support the present study.
Response 6: According to the reviewers' opinions, we have supplemented the references during the discussion.
Point 7: Conclusion: needs to be precise and highlights only the major findings of the study.
Response 7: There are some things in the conclusion that do not need to be emphasized. I have deleted them. The details are as follows:
This study demonstrated the positive effects of alfalfa physiological and biochemical indicators and anatomy on alfalfa photosynthesis, the synergistic improvement in photosynthetic efficiency and the photosynthetic area of alfalfa fertilization treatment further contributed to the improvement of alfalfa production performance. We found that Tr(r=0.519) and Pn(r=0.351) had a greater direct impact on the formation of dry matter yield, while Chl(r=0.763) and ST(r=0.749) had the greatest comprehensive impact. The increased application of N and P fertilizers increased the AQY of alfalfa leaves, the Pmax and LSP of alfalfa leaves under N application treatment were significantly higher than those of the non-N treatment. Under the influence of long-term external fertilization, the morphological structure of the leaves undergoes certain physiological adaptations. For example, N and P increase the thickness of the ST of alfalfa, which will facilitate the entry and exit of gas and water, and will further affect photosynthetic indices, such as stomatal conductance and transpiration rate of alfalfa leaves. Increased PT thickness will also enhance the adaptability of plant leaves to strong light, and then increase the Pmax and LSP. These results indicated that the application of N and P fertilizers contributed to the adaptation of alfalfa leaf anatomy and the improvement of photosynthetic capacity, which was beneficial to the improvement of alfalfa DMY, growth traits and nutritional quality, with the most obvious effect at 120 kg·ha-1 of N and 100 kg·ha-1 of P2O5 application.
